# HLA3DB: comprehensive annotation of peptide/HLA complexes enables blind structure prediction of T cell epitopes

Sagar Gupta [1,2,4], Santrupti Nerli[1,4], Sreeja Kutti Kandy[1], Glenn L. Mersky[1] & Nikolaos G. Sgourakis [1,3]

The class I proteins of the major histocompatibility complex (MHC-I) display epitopic peptides derived from endogenous proteins on the cell surface for immune surveillance. Accurate modeling of peptides bound to the human MHC, HLA, has been mired by conformational diversity of the central peptide residues, which are critical for recognition by T cell receptors. Here, analysis of X-ray crystal structures within our curated database (HLA3DB) shows that pHLA complexes encompassing multiple HLA allotypes present a discrete set of peptide backbone conformations. Leveraging these backbones, we employ a regression model trained on terms of a physically relevant energy function to develop a comparative modeling approach for nonamer pHLA structures named RepPred. Our method outperforms the top pHLA modeling approach by up to 19% in structural accuracy, and consistently predicts blind targets not included in our training set. Insights from our work may be applied towards predicting antigen immunogenicity, and receptor cross-reactivity.

The class I major histocompatibility complex (MHC-I) protein presents self, foreign, or aberrant peptides to provide a mechanism of immune surveillance via CD8[+] cytotoxic T cell lymphocytes[1]. The human version of MHC-I, Human Leukocyte Antigen class I (HLA-I), is encoded by the highly polymorphic *HLA* gene locus[2]. There are three classical (*HLA-A*, *HLA-B*, and *HLA-C*) as well as three non-classical *HLA* genes (*HLA-E*, *HLA-F*, and *HLA-G*) that combine to encode over 35,000 HLA allotypes[3,4]. Each allotype can display a repertoire of up to $10^8$ epitopic peptide sequences, defining distinct peptide/HLA-I (pHLA) complexes[5]. The large HLA sequence variability and resulting differences in peptide repertoires can cause alternative disease susceptibility[6]. Additionally, this diversity across allotypes ensures species viability as all antigens can likely be presented by at least one allotype in the population. At the structural level, peptides of length 8 to 15 amino acids bind to the HLA groove, which consists of six major pockets labeled from A to F[7]. Canonical binding occurs via stable anchoring interactions within the B and F pockets of the HLA and the second (P2) and last (PΩ) peptide

residues, respectively, with longer peptides bulging out of the groove[8]. Peptide binding specificity is restricted by allotype because the amino acids of peptide anchor residues complement those in the HLA groove, leading to allele-specific sequence motifs. Meanwhile, the amino acid propensities of non-anchor positions are generally more permissive, which, when combined with the HLA's polymorphic nature, leads to a large combinatorial complexity in the number of potential pHLA structures. As a result, estimating pHLA binding affinities has been resolved using sequence data alone[9]. Predicting peptide immunogenicity and cross-reactivity, on the other hand, necessitates detailed structural information, which has been more challenging to model due to subtle structural changes which may arise from amino acid variations[10].

The high degree of structural conservation among pHLA complexes suggests that structure-based modeling of novel peptide/HLA antigens can be addressed using conventional comparative modeling approaches[11,12]. Following the first crystallographic structure

[1]Center for Computational and Genomic Medicine, Department of Pathology and Laboratory Medicine, Children's Hospital of Philadelphia, Philadelphia, PA, USA. [2]College of Arts and Sciences, University of Pennsylvania, Philadelphia, PA, USA. [3]Department of Biochemistry and Biophysics, Perelman School of Medicine, University of Pennsylvania, Philadelphia, PA, USA. [4]These authors contributed equally: Sagar Gupta, Santrupti Nerli.
✉ e-mail: nikolaos.sgourakis@pennmedicine.upenn.edu

determination of a peptide/HLA complex[13], we now have access to a substantial dataset of reliable structures[14]. Recent efforts in de novo modeling of the pHLA complex have focused on three main docking strategies: constrained backbone, constrained termini, and incremental peptide reconstruction[15]. Constrained backbone based predictions assume that all backbones of the same length are similar in their conformation[12,16–21]. On the other hand, constrained termini methods offer a less restrictive view as they assume that termini residues are confined to defined pockets in the peptide-binding groove, while the locations of the other peptide residues are said to be variable[22–27]. Lastly, fragment-based docking strategies sample the peptide backbone ab initio, allowing for a greater degree of flexibility[28,29]. In all three cases, the modeled structures are ranked and compared based on either peptide backbone Root Mean Square Deviation (RMSD) or all-atom RMSD. While these methods have generally been able to accurately model the HLA groove and N- and C-termini of the peptide, they often miss critical details in the center of the peptide. The sequence and conformational diversity offered by the central region of the peptide defines the immunologically important area of pHLA structures[30–32]. Thus, accurate modeling of the center of the peptide is difficult but necessary to understand and predict the molecular basis for immunogenicity.

Here, we develop a database of pHLA structures (HLA3DB) and classify peptide backbones based on the sequence separation between their primary anchor residues. Focusing on nonamer peptides, we apply an internal coordinate-based system to compare backbones in dihedral angle space, and find that HLA-I peptides can sample a discrete set of conformations. Additionally, we observe that similar backbones can be obtained despite dissimilar peptide and HLA sequences, revealing the complexity of convergent interactions which ultimately define the peptide backbone. Exhaustive modeling simulations using Rosetta reveal that distinct peptide backbones necessitate unique biases of central peptide sequences arising from several factors, including steric hindrance. Finally, we combine our basis set of distinct structural templates with a regression model trained on a physically relevant energy function to develop a structural modeling approach for nonamer/HLA complexes (RepPred). Using a cross-validation benchmark, we find that our method outperforms six state-of-the-art methods[24,25,29,33–35], showing a 19% improvement in accuracy relative to the top method[35], while consistently identifying the correct templates for target backbones that are sparsely populated in the Protein Data Bank (PDB). Additionally, independent testing using a blind set of targets shows comparable accuracy to our benchmarking results. Our findings enable accurate modeling of peptide/HLA structures at scale, paving the way for accurate prediction of peptide immunogenicity and cross-reactivity[10,36–41].

## Results

### Analysis of pHLA structures uncovers the basis for peptide conformational diversity

A comprehensive structural analysis of pHLA complexes requires a curated database of high-resolution X-ray structures. While there are publicly available databases that store MHC-I structural data[42–50], they do not provide a consistent format needed for further automated analysis. Using the RCSB PDB Search API[51], we developed an automatic protocol for extracting and annotating pHLA structures with peptide lengths from 8 to 10 residues (Supplementary Fig. 1 and Methods). From each PDB entry, we retained the $\alpha_1$ and $\alpha_2$ domains of the MHC heavy chain, referred to as the MHC platform, in addition to the peptide. Leveraging the IPD-IMGT/HLA Database[4] as a reference, we then assigned an HLA allotype to each human MHC-I X-ray structure. The resulting set of curated pHLA platform structures were stored in HLA3DB (https://hla3db.research.chop.edu), a publicly available, auto-updating database. HLA3DB consists of 393 structures with 15 octamer, 296 nonamer, and 82 decamer peptides (Fig. 1a). In terms of HLA

peptide binding specificities, the classical HLA-Ia allotypes present in the database cover five out of six known HLA-A supertypes and all HLA-B supertypes[52], in addition to two non-classical type Ib complexes (HLA-E and HLA-G). As expected, the A02 supertype is represented by 42% of the structures in the dataset. Notwithstanding, the wide range of HLA groove and peptide sequences present in our dataset provides a comprehensive sampling of the conformational space covering possible peptide backbones to guide structural modeling efforts.

The prominent peptide classification scheme for immunopeptidomics analysis utilizes a linear view of epitope lengths, where for most HLA allotypes, the peptide's second (P2) and last (PΩ) position bind to the B and F pockets of the peptide-binding groove, respectively. However, notable exceptions exist, including HLA-A*02:01 bound to the MART-1 nonamer antigen (LAGIGILTV), which has non-canonical anchor residues at P1 and PΩ (Supplementary Fig. 2a). This exception has important ramifications from a structural perspective, indicating that peptide configurations of the same length can deviate significantly. Thus, to classify peptides of different lengths according to a global frame of reference defined by the HLA groove, we employed an anchor residue-based scheme in which an anchor class is defined as the sequence separation between the two most distant anchor residues (Fig. 1b and Methods). Using this scheme, we categorized all pHLA structures in HLA3DB and found three distinct classes of peptides: Δ6, Δ7, and Δ8. Generally, octamer, nonamer, and decamer peptides reside in the Δ6, Δ7, and Δ8 classes, respectively, but this is not always the case (Supplementary Fig. 2b). As expected, the Δ7 anchor class contains the most structures since nonameric peptides are the dominant sequence length presented by MHC-I molecules[53].

Next, we evaluated the distance between the Cα atoms of the anchor residues for each anchor class, termed the anchor distance. Despite differences in peptide-binding motifs across HLA supertypes, we find that the anchor distance is heavily confined by the geometry of the HLA groove. Examination of pHLA structures reveals overlapping distributions of anchor distances, with a median of 18 Å, 18.5 Å, and 19 Å, for the Δ6, Δ7, and Δ8 classes, respectively (Fig. 1c). As the distance separation between anchor residues increases, the peptide's central bulge becomes more pronounced, resulting in a more concave structure. A small set of outliers can be attained through either β-strand extended (Supplementary Fig. 2c, d, f) or α-helical condensed peptide backbone conformations (Supplementary Fig. 2e). In general, peptides belonging to the same anchor class show a relatively restricted distance distribution, supporting a theme where divergent conformations are attained through local changes in internal backbone degrees of freedom, rather than through global changes in the overall length of the displayed peptide antigen.

We next sought to characterize the conformational diversity of peptides belonging to the same overall anchor class. Thus, we initially focused on Δ7 peptides ($n = 303$) as they were the most common classification in our dataset. While these peptides exhibited a narrow anchor distance distribution from 17.5 Å to 20 Å, highly divergent conformations were observed, which could influence recognition by T-cell receptors (TCRs). Building on the knowledge that TCRs preferentially interact with the central bulge of the peptide, we employed a structural framework, termed the fixed-local frame, to capture peptide conformational diversity (Fig. 1d). We established that the primary anchor residues (P2 and P9) provide a fixed frame of reference by defining a narrow range of allowed Cα-Cα distances. Concurrently, the central portion of the peptide, specified as positions 4 to 7, define peptide backbone diversity through changes in the φ and ψ dihedral angles.

To further highlight that dihedral angle variations occurring at specific peptide positions are required to fit a nonamer in the peptide-binding groove, we modeled different regular backbone structures de novo in the absence of the MHC and measured the resulting anchor distance (Methods). Peptide anchor distances within the observed

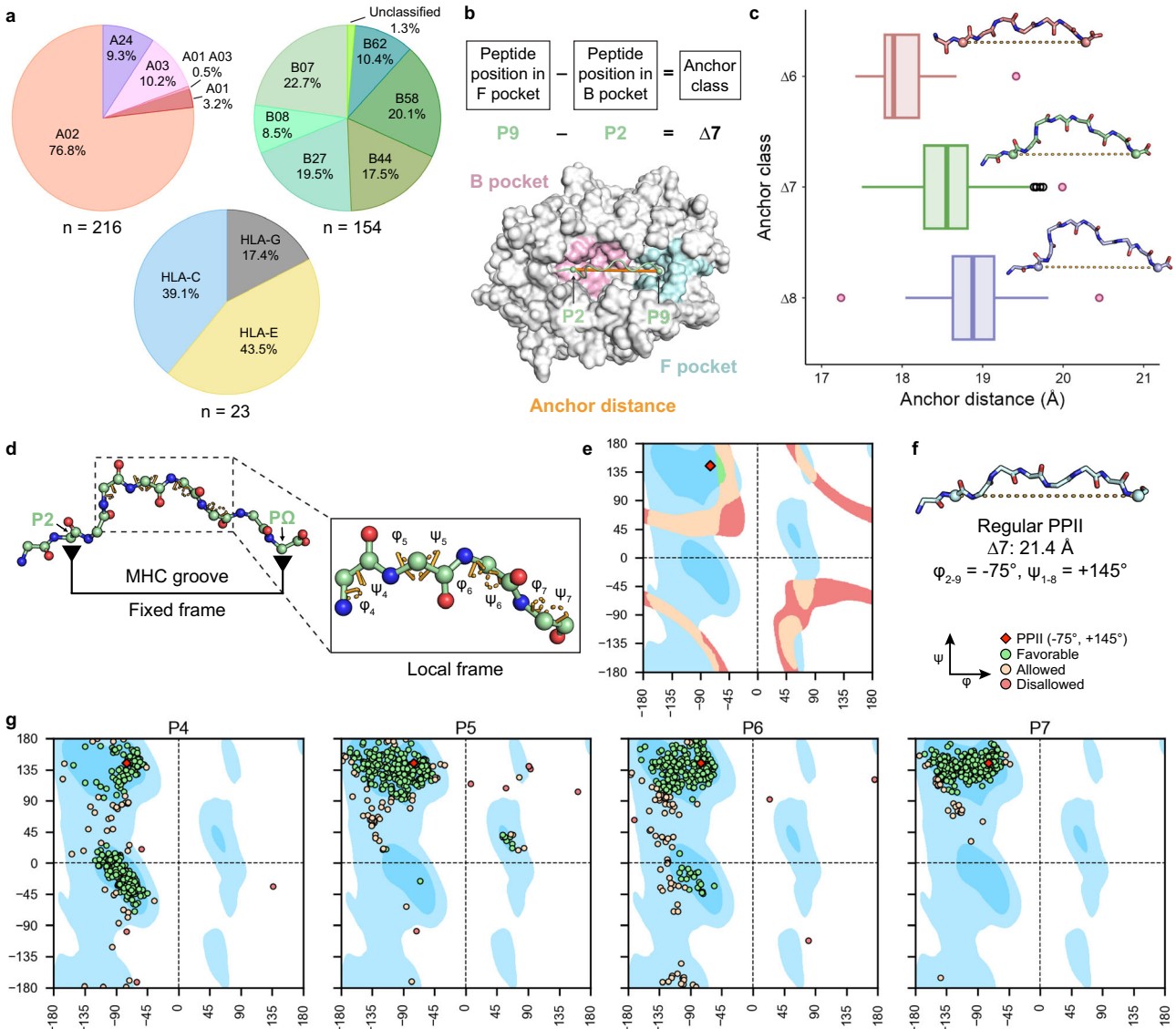

**Fig. 1 | Analysis of pHLA structures in HLA3DB uncovers the basis for peptide conformational diversity. a** Distribution of pHLA structures in HLA3DB by supertype for HLA-A and HLA-B and single-resolution allotype for HLA-C, HLA-E, and HLA-G. **b** Schematic depicting how anchor class and anchor distance are defined using the Δ7 anchor class as an example. Cα atoms of anchor residues are shown as green spheres and connected by an orange solid line depicting the anchor distance. B and F pockets are shaded in pink and blue, respectively, and the remainder of the HLA is colored in gray. **c** Distribution of anchor distances for each anchor class with the center indicating the median (Δ6 pHLA structures: $n = 15$; Δ7 pHLA structures: $n = 303$; Δ8 pHLA structures: $n = 75$). Whiskers extend to the furthest values that lie within the 75th and 25th percentile value ± 1.5 times the interquartile range. Outliers are shown in black and pink circles with pink data points elaborated in Supplementary Fig. 1c–f. Peptide backbones corresponding to the median anchor distance of each anchor class are shown above each respective boxplot (Δ6: PDB ID 1E28 [https://doi.org/10.2210/pdb1E28/pdb], Δ7: PDB ID 1K5N [https://doi.org/10.2210/pdb1K5N/pdb], Δ8: PDB ID 3I6K [https://doi.org/10.2210/pdb3I6K/pdb]). **d** A schematic of the fixed-local framework for conformational diversity. On the right, the inset highlights the central bulge of the peptide and the respective dihedral angles in orange dashed sectors. Backbone heavy atoms are shown as spheres. **e** General Ramachandran plot showing dihedral angle pairs satisfying the anchor distance distribution seen in the Δ7 anchor class. Favorable overlaps are colored green, allowed overlaps are colored cream, and disallowed overlaps are colored red. Favorable regions are shaded blue, allowed regions are shaded light blue, and unfavorable regions are shaded white. PPII conformation is shown as a red diamond at (−75°, +145°). **f** Backbone of a de novo designed poly-glycine nonamer with all dihedral angles set to the PPII angle (−75°, +145°). Cα atoms of anchor positions are shown as spheres and connected via an orange dotted line to indicate anchor distance. **g** General Ramachandran plots of all Δ7 peptides ($n = 303$). Plots are shaded identically to (**b**) with discrete points instead of shaded surfaces.

range for Δ7 pHLA structures (17.5 Å to 20.0 Å) were plotted as continuous surfaces on a Ramachandran plot (Fig. 1e). We found a single overlapping area of this surface with the favorable region of a Ramachandran plot that was adjacent to the ideal polyproline type II (PPII) helical conformation (φ = −75°, ψ = +145°)[54]. However, the anchor distance of an ideal PPII peptide was found to be 21.4 Å, lying outside of the distribution observed for the Δ7 anchor class (Fig. 1f). Notably, even pHLA complexes with a near regular PPII peptide conformation required slight variations to allow for an acceptable anchor distance

distribution (Supplementary Fig. 3). This calculation reveals that, in order to satisfy the observed range of anchor distances in the Δ7 peptides, individual residues within Δ7 nonamers must adopt configurations that deviate from the ideal PPII conformation, introducing structural diversity.

We next analyzed the distributions of φ and ψ dihedral angles for each peptide position among peptides in the three anchor classes (Fig. 1g, Supplementary Figs. 4 and 5). For the majority of structures, the canonical anchor residues (P2 and PΩ) and adjacent residues were

clustered around the PPII conformation. For instance, in Δ7 peptides, P2, P3, and P8 were conserved. On the other hand, residues in the center of the peptide showed frequent deviations from the PPII region towards a more extended (β-strand) or more condensed (α-helix) structure. The magnitude and order of these divergences, driven by the identity of both the HLA groove and peptide residues, defines the conformational diversity of epitope backbones. These results support that the conserved anchor residues (P2 and PΩ), which are primarily involved in HLA binding, display little variability from the ideal PPII conformation, while changes in the central part of the peptide promote conformational diversity necessary for specific TCR recognition.

## Unbiased classification reveals conserved peptide backbones across HLA allotypes

Guided by insights from our analysis of peptide dihedral angles, we apply an existing internal coordinate-based metric[55] to measure structural divergence by comparing the φ and ψ dihedral angles of the center of the peptide, termed the D-score (Methods). This is in contrast to RMSD, which is defined in Cartesian space using a global frame of reference[56]. To compare the performance of D-score vs RMSD in comparing peptide backbone conformations, we focused on Δ7 peptides as they are the most common in our dataset and determined the backbone heavy atom RMSD of P4 to P7 and the D-score for each pair of structures. While low D-score values generally corresponded to low RMSD values, there were significant differences between peptide backbones leading to an increased D-score that was not captured by RMSD (Supplementary Fig. 6a). Thus, our results suggest that D-score more accurately captures the difference in backbone configuration between two peptides. In contrast, the global frame used by RMSD does not accurately assess variations between individual φ and ψ dihedral angles, limiting a more precise, quantitative analysis. We define the D-score criteria as a D-score of less than 1.5 as this ensure that the RMSD is less than 1 Å between backbones. We scaled the D-score for Δ6 and Δ8 peptides according to the number of angles included in the summation. An exemplar comparison of two structures reveals how significant dihedral angle differences cannot be captured by backbone heavy atom RMSD, leading to an inaccurate evaluation of structural similarity (Supplementary Fig. 6b). We find that a large dihedral angle difference at P5 can cause a substantial backbone deviation at P6 and P7 (Supplementary Fig. 6c). Nonetheless, the resulting RMSD is less than 1 Å, suggesting a near identical peptide configuration. These results establish that the D-score, a metric in an internal coordinate system, is a more accurate measure than RMSD in determining the structural similarity of the central part of peptide backbones.

To assess the extent of structural similarity in our dataset, we determined the number of neighbors (non-self peptides which satisfy the D-score criteria) for each Δ7 peptide. Overall, each structure had a median of 28 neighbors; however, the number of similar backbones ranged from 0 to over 100 (Supplementary Fig. 7a). Furthermore, we generally found that two pHLA complexes could be neighbors irrespective of their peptide or allele sequence, signifying that interactions between divergent peptide and HLA residues can yield similar backbone structures. For instance, one structure (PDB ID 5VGD [https://doi.org/10.2210/pdb5VGD/pdb]) had 12 neighbors, each of which exhibited similar peptide backbone dihedral angles to 5VGD, despite their allotype (Supplementary Fig. 7b–d). As a result, this finding confirmed that HLA allotype, which restricts the peptide sequence at anchor residues, plays a negligible role in defining the backbone conformation, and instead structural diversity is defined by non-anchor residues, specifically P4 to P7 for Δ7 peptides.

Recognizing that the backbones in our dataset can adopt recurrent structures, we sought to establish a minimal set of peptide configurations which could describe the entire conformational space for peptides in each anchor class. Due to its limited size, our dataset does not accurately capture the frequency of different backbones which can be adopted by immunodominant peptide antigens. Thus, it was critical that peptide conformations with zero neighbors were retained. Hence, we chose a greedy algorithm based on the fixed-local frame and constructed a binary symmetric adjacency matrix, using the D-score to measure structural similarity (Fig. 2a and Methods). In this process, the structure with the most neighbors, known as a discrete peptide backbone, was identified and it, along with its neighbors, was removed from the matrix. This procedure was continued iteratively until the matrix was empty, i.e., until all structures were accounted for. In this way, we were able to capture the full peptide conformational space without redundancy while also accounting for underrepresented backbones. As per our expectation, we found that our set of 303 Δ7 peptides could be represented by a minimal set of 35 discrete peptide backbones (Supplementary Fig. 8). Similarly, Δ6 ($n = 15$) and Δ8 ($n = 75$) peptides could be represented by 3 and 34 backbones, respectively (Supplementary Figs. 9, 10 and Supplementary Table 1).

We focused further analysis on Δ7 pHLA structures as they were the most common in our dataset. To visualize the extent to which our discrete peptide backbones spanned the peptide conformational landscape, we used a two-dimensional PCA plot. This revealed that discrete peptides were not only able to capture common conformations such as those located in the lower left of the PCA plot, but also rare configurations represented by data points in the upper center region of the plot (Fig. 2b). Thus, our greedy algorithm successfully captured the entire conformational space. A historical analysis of Δ7 pHLA structures in the dataset showed a geometric progression in the number of structures deposited (Fig. 2c). Notably, the rise in the number of discrete peptide backbones has comparatively stagnated over the past 5 years. We next sought to understand the supertype distribution and number of neighbors for each discrete epitope. We defined a discrete peptide backbone and its neighbors as a distinct set of backbones. Since supertypes were generally diverse among each set of backbones, we recapitulated our finding that similar epitope conformations are not solely defined by the groove residues, but rather a convergent set of interactions between HLA and peptide residues (Fig. 2d). Furthermore, the most common class (Δ7-1_6J1V), accounted for 35% of all pHLAs in our dataset. In contrast, just under half of the discrete peptides defined unique classifications with just one neighbor. These results indicate that our basis set of 35 discrete peptide backbones can capture the diversity of pHLA structures in our dataset and that the existing structural data likely cover the majority of possible Δ7 peptide backbone conformations.

## Exhaustive enumeration of the peptide sequence space reveals structural biases

Next, we sought to investigate if peptide sequence biases existed across distinct backbones. However, our dataset dwarfed the potential sequence space of $20^9$ peptides, containing just 247 nonredundant Δ7 nonamer sequences. Thus, we expanded our existing peptide sequence space coverage using a fixed backbone Rosetta design[57,58] technique (Fig. 3a and Methods). To ensure that the peptide-binding groove was not a confounding variable, we restricted our analysis to HLA-A*02:01, the most common allotype in our dataset. Using the aforementioned greedy algorithm, we determined that there were 20 distinct peptide backbones across all 121 HLA-A*02:01 Δ7 nonamers. We focused our analysis on the five most common HLA-A*02:01 configurations, determined by their number of neighboring conformations. Next, we narrowed down the set of potential peptide sequences by applying two constraints. First, along with the center of the peptide (P4 to P7), P3 and P8 were also included to capture any effects that adjacent positions may have on the peptide sequence of the central residues. Second, we used a previously published matrix[59], which reports amino acid preferences on a per-position basis to narrow our sequence space for P3 to P8 from 20 amino acids to 8 to 11 amino acids,

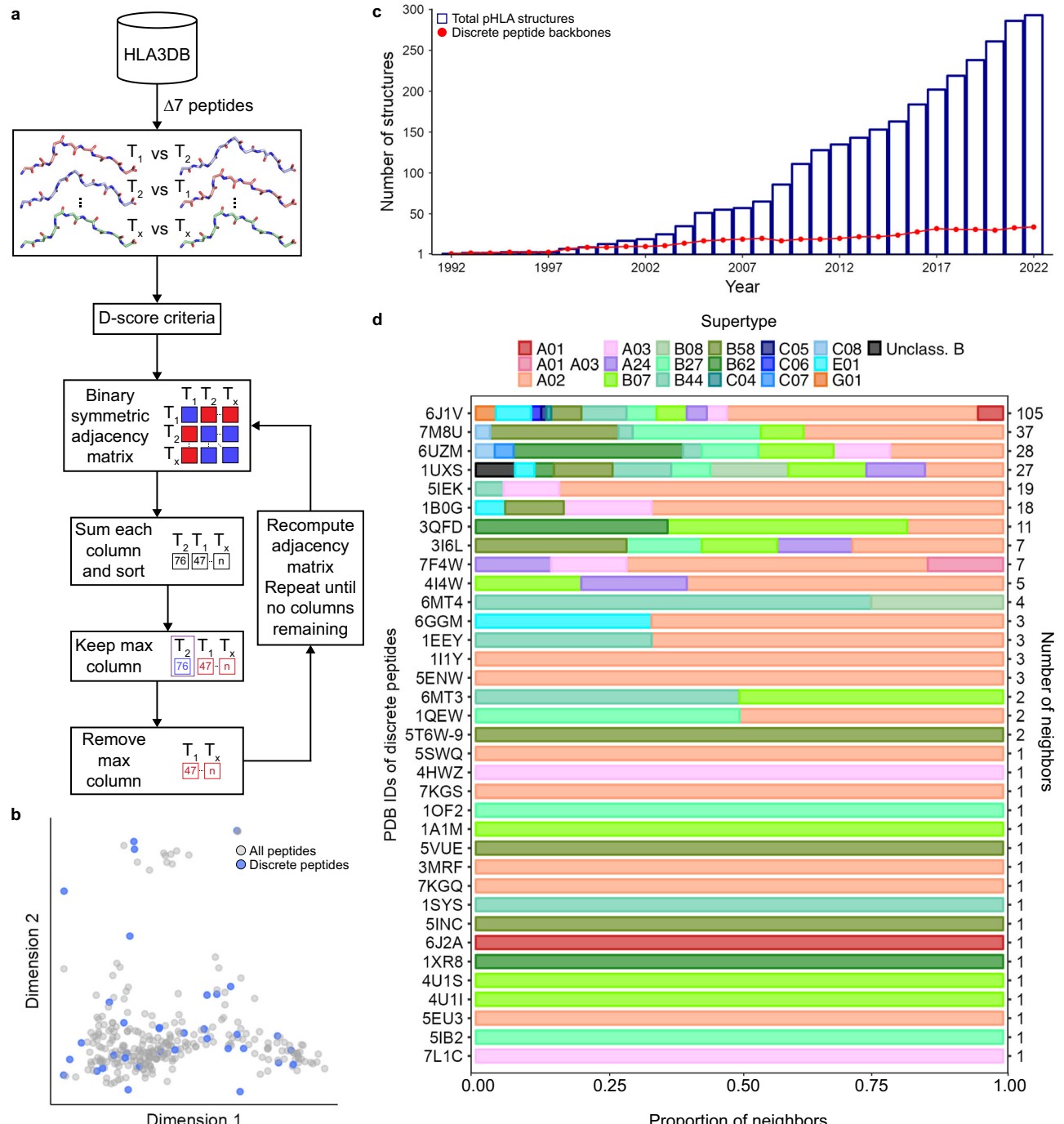

**Fig. 2 | Unbiased classification reveals conserved peptide backbones across HLA allotypes. a** A schematic of the greedy algorithm used to select discrete peptide backbones. **b** A two-dimensional PCA plot standardized with the sine of the dihedral angles of P4 to P7 explaining 66% of the variance. **c** Historical analysis of the cumulative number of Δ7 structures (blue) and discrete peptide backbones (red) in HLA3DB as determined iteratively by the greedy algorithm. **d** The proportion of supertype in each discrete peptide's set of neighbors. The number of neighbors for each discrete peptide is shown in the right y-axis.

depending on the position (Supplementary Table 2). We also considered proline at each position due to its restraining effects on the peptide backbone. In total, we evaluated 784,080 peptide sequences for each of the five distinct peptide backbones. Iterating through the set of peptide sequences, we generated models, determined the total score of each model using the ref2015 Rosetta energy function[60], and only considered the peptide sequences of models that were in the top one percent of the energy distribution, i.e., 7840 models. While ref2015 has struggled with capturing subtle changes involved electrostatic interactions[61], prior research suggests its potency in capturing steric

hindrance[58]. Thus, we use the energy function along with practical constraints on the peptide sequence applied by the groove, to enable exhaustive structural modeling to substantially expand our dataset of HLA-A*02:01 epitopes.

To determine if these backbone-specific peptide sequences were distinct from the baseline peptide binding preferences, we computed the Kullback–Leibler (KL) divergence[62] at each position using 32,483 peptides experimentally verified to bind to HLA-A*02:01. We utilized peptide binding data from HLA-A*68:01, an allele belonging to the A02 supertype and therefore showing a similar peptide binding specificity,

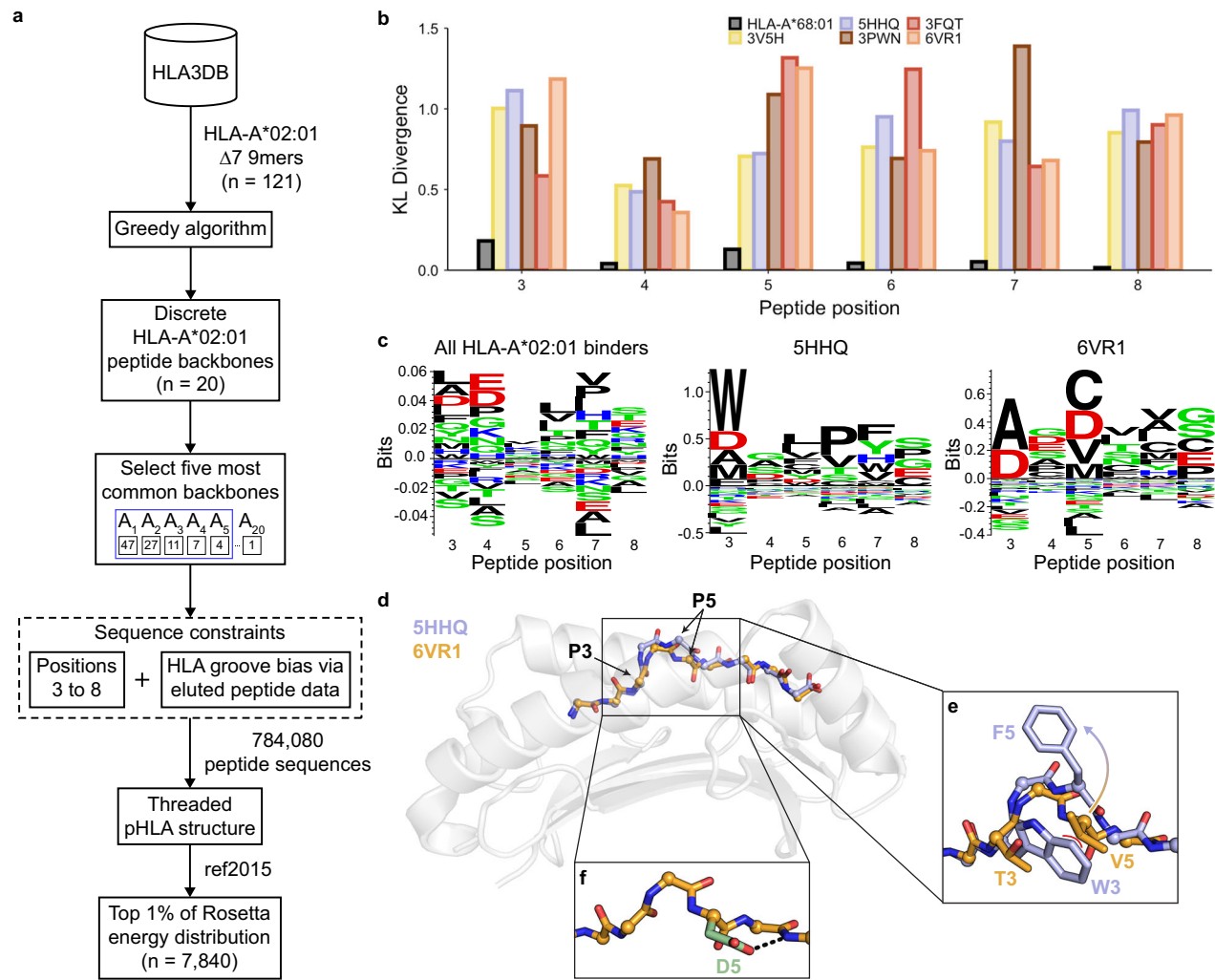

**Fig. 3 | Exhaustive enumeration of peptide sequence space reveals biases imposed by different backbone conformations. a** A schematic of the exhaustive structural modeling conducted to augment the existing peptide sequence space for HLA-A*02:01. **b** Kullback−Leibler (KL) divergence between each of the predicted sequence space of the five most common HLA-A*02:01 backbones represented by PDB IDs and HLA-A*02:01 reference sequence space. A comparison between the eluted peptides bound to HLA-A*68:01 and those bound to HLA-A*02:01 was included as a negative control. **c** Peptide sequence logos of all eluted peptides shown to bind to HLA-A*02:01 via the IEDB ($n = 32{,}483$) and from structural modeling results ($n = 7840$). Created using Seq2Logo[90]. **d** Structural overlay of the peptide backbones (5HHQ [https://doi.org/10.2210/pdb5HHQ/pdb], blue and 6VR1 [https://doi.org/10.2210/pdb6VR1/pdb], orange) bound to HLA-A*02:01 (gray) with the peptide Cα atoms shown as spheres. **e** An enlarged view of the middle of the peptide with side chains depicted for P3 and P5. Coloring of the peptide backbone and side chain is identical to (**d**). Steric hindrance is shown as a red curve. **f** Structural model of the 6VR1 [https://doi.org/10.2210/pdb6VR1/pdb] peptide backbone with P5 mutated to aspartate, as shown in green. A hydrogen bond is shown as a dashed black line.

as a measure of the lower bound of the KL divergence. Our results revealed that all five distinct backbones exhibited peptide sequence biases (Fig. 3b). While the KL divergence was consistently lower for P4 compared to other positions, the HLA-A*02:01 binding motif contained a known bias[63] for negatively charged amino acids at P4, ultimately decreasing the perceived sequence bias. We confirmed the quantitative values given by the KL divergence by visually comparing the sequence logo of all known HLA-A*02:01 binders to those obtained by exhaustive structural modeling (Fig. 3c and Supplementary Fig. 11a−c).

We next sought to explain why different backbones could lead to distinct peptide sequence biases. As an example, we chose two peptide configurations (PDB IDs 5HHQ [https://doi.org/10.2210/pdb5HHQ/pdb] and 6VR1 [https://doi.org/10.2210/pdb6VR1/pdb]) with a D-score of 5.9 (Fig. 3d). A per-position D-score analysis revealed that the deviations occurred at P5 and P6 where the backbones diverge (Supplementary Fig. 11d). Peptide sequence logos of these backbones also revealed significant differences. For 5HHQ's

backbone, our modeling approach reported a preference for tryptophan at P3, which was also found in the native peptide sequence (GI**W**GFVFTL). Meanwhile, the sequence logo of 6VR1 favored small amino acids at P3 such as alanine since a bulky group at this position would clash with most side chains at P5 (Fig. 3e). The native peptide sequence of 6VR1, HM**T**EVVRRC, followed this trend as well. A backbone deviation in 5HHQ diverted the side chain at P5 away from the tryptophan, avoiding an energetically unfavorable steric hindrance issue and encouraging a broad selection of sequences. Additionally, we found a preference for aspartate at P5 in the sequence logo of 6VR1, which can be explained by a favorable hydrogen bonding interaction occurring between the side chain and backbone (Fig. 3f). Due to 5HHQ's backbone conformation, the side chain of P5 cannot contact the backbone nitrogen atom of P7 and thus does not display the same sequence bias. This comparison demonstrates how the backbone conformation could impose local steric constraints or introduce favorable interactions through crosstalk between peptide backbone and side chain features.

## A trained regression function allows for accurate structural modeling

We aimed to harness the knowledge that all pHLA structures in our dataset could be captured by a set of discrete peptide backbones to enable accurate structural modeling of new antigens on all HLA allotypes. To this end, we developed RepPred, an automated modeling method which utilizes discrete backbones as templates for homology modeling of Δ7 nonamer epitopes (Fig. 4a). Briefly, a target peptide sequence is threaded onto 33 stable discrete backbone templates, followed by structural refinement in Rosetta to create an initial set of models. We computed the per-residue Rosetta energy terms of the models[29] and utilized these values as input features for a support vector machine regression (SVR) function, enabling the prediction of the D-score between the native crystal structure and the model. RepPred reports the best model as that with the lowest predicted D-score.

To assess the accuracy of RepPred, we perform a benchmark on all targets in our dataset against a set of nonhomologous discrete peptide backbones. Thus, we evaluate a total of 7775 target-template pairs in a leave-one-out cross-validated manner whereby the models of the target structure are removed from the training set. As expected, we observe that 96% of models have a sub-angstrom HLA backbone RMSD (Supplementary Fig. 12a). Recognizing the overrepresentation of HLA-A*02:01 (A02) structures in our dataset, we first explore the accuracy for A02 targets separately from all other allotypes, i.e., non-A02, targets. We define a successful model as one that has a D-score less than 1.5 relative to its known crystal structure. RepPred reports a structurally accurate model for 63% of A02 targets ($n = 102$) and, on the basis of backbone RMSD for the middle of the peptide, it selects a sub-angstrom model for 92% of A02 targets (Fig. 4b). While the refinement step can cause the backbone to drift from the template conformation, selecting the best model for A02 targets by D-score gives an accuracy of 99% which is generally in agreement with the finding that the representatives cover the conformational space. On the other hand, a random selection of the representative backbone rarely (10%) selects

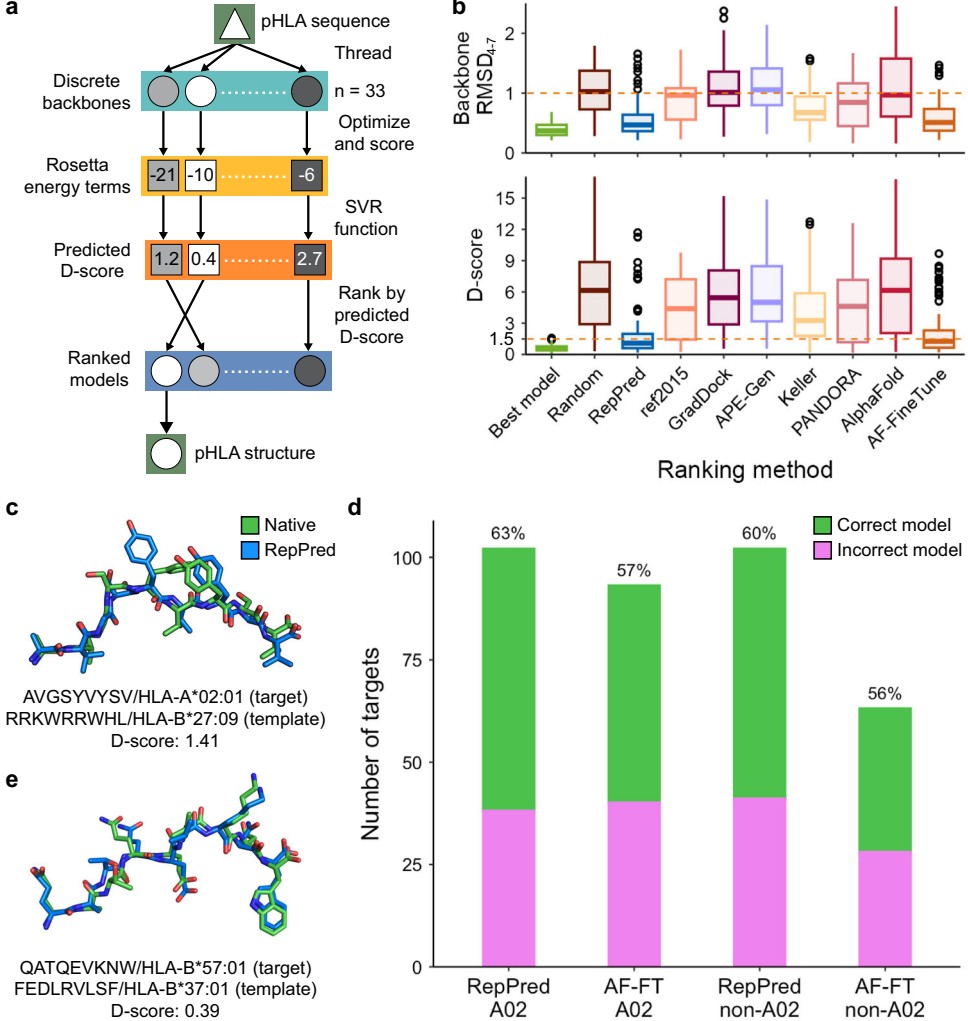

**Fig. 4 | Structural modeling of Δ7 nonamer/HLA complexes (RepPred) and comparison to state-of-the-art methods. a** Full workflow of the RepPred structural modeling method. **b** Boxplots showing the distribution of the D-score and peptide backbone heavy atom RMSD of RepPred and six state-of-the-art methods, sorted by publication date, for A02 targets with the center indicating the median (Best model, RepPred, ref2015: $n = 102$; GradDock: $n = 90$; APE-Gen: $n = 99$; Keller: $n = 101$; PANDORA: $n = 100$; AlphaFold: $n = 101$; AF-FineTune: $n = 93$). Whiskers extend to the furthest values that lie within the 75th and 25th percentile value ± 1.5 times the interquartile range and outliers are shown in black circles. An orange dashed line is at a D-score of 1.5 and an RMSD of 1.0. **c** Structural superposition of

the RepPred model (template PDB ID 1OF2 [https://doi.org/10.2210/pdb1OF2/pdb]) and native structure (PDB ID 7U21 [https://doi.org/10.2210/pdb7U21/pdb]) of the blind target melanoma antigen AVGSYVYSV bound to HLA-A*02:01. The target sequence and allotype is written below the structure followed by the sequence and allotype of the template. **d** Comparison of RepPred against AF-FT for A02 and non-A02 targets. Accuracy is shown as a percentage over each bar. **e** Structural superposition of the RepPred model (template PDB ID 6MT6 [https://doi.org/10.2210/pdb6MT6/pdb]) and native structure (PDB ID 7R7Y [https://doi.org/10.2210/pdb7R7Y/pdb]) of the blind target HIV antigen QATQEVKNW bound to HLA-B*57:01. Formatting is identical to (**c**).

the correct decoy, as expected. To further validate our modeling approach, we conduct a blind test by assessing the accuracy of RepPred on five Δ7 nonamer/HLA-A*02:01 structures in HLA3DB deposited after the cutoff date for our dataset. In line with the benchmark results, we accurately predict three out of five structures within a D-score of 1.5, allowing for accurate placement of side chains in their native rotameric states (Supplementary Fig. 12b). Our blind test included a melanoma neoantigen (AVGSYVYSV)[64] which we modeled with a template structure originating from HLA-B*27:09, an allele not found in the target's supertype family (Fig. 4c). This finding recapitulates the trend that similar backbones can occur independent of allele identity, allowing us to use all observed backbones as a basis set for structural modeling.

Next, we compare RepPred with six existing approaches for pHLA-I structural modeling: GradDock[24], APE-Gen[33], a method by ref. 29 that we refer to as Keller, PANDORA[25], AlphaFold2[34], and a peptide/MHC fine-tuned version of AlphaFold (AF-FT)[35]. We focus our comparison on A02 targets and find that RepPred outperforms five of the six methods by at least 32% with respect to D-score (Fig. 4b). Additionally, RepPred performs competitively (62% ≤ 1.5 Å) when considering all-atom RMSD for the middle of the peptide (Supplementary Fig. 12c). However, we found that nearly half of the A02 targets are neighbors to a discrete backbone (PDB ID 6J1V [https://doi.org/10.2210/pdb6J1V/pdb]). Hence, a naive method could achieve 50% accuracy by simply adopting this backbone conformation for all targets. While RepPred does achieve higher accuracy (76%) for the most common backbone classification, it maintains 50% accuracy for all other backbones, resulting in an overall accuracy of 63% (Supplementary Fig. 12d). Meanwhile, AF-FT reports an overall accuracy of 57% ($n = 93$) and an accuracy of 78% and 38% for the most common and all other backbones, respectively. Thus, while AF-FT is influenced by the bias of the most common configuration, in utilizing discrete peptide backbones, RepPred weighs each conformation equally and reduces the effect of this bias. As a result, RepPred shows a slight improvement across all backbones and a notable enhancement for less common conformations. We next sought to conduct a similar comparison of RepPred to AF-FT for non-A02 targets. RepPred achieves 60% accuracy for these structures ($n = 102$) (Fig. 4d), an accuracy comparable to that of A02 targets. We further validated our method through a blind test of seven non-A02 targets in which we obtained an accuracy of 57% (Supplementary Fig. 12b). RepPred models an immunodominant HIV epitope (QATQEVKNW)[65] with near-native side chain placement as a result of high-fidelity generation of the peptide backbone (Fig. 4e). A comparison to AF-FT reveals an overall accuracy of 56% for non-A02 targets ($n = 63$). When assessing performance for both methods based on backbone conformation, we observe that RepPred performs 19% better than AF-FT for common backbones and comparably for all other conformations (Supplementary Fig. 12e). Taken together, RepPred models nonamer pHLA complexes with high accuracy, a finding which resurfaces in blind tests of immunologically relevant antigens and performs better than state-of-the-art methods.

## Discussion

Our results characterize peptide backbone diversity across all pHLA-I structures and builds on the work of Dunbrack and colleagues who clustered antibody and kinase conformations[66–70]. Using HLA3DB, our database of peptide/HLA structures, we broadly categorize peptides using our anchor classification scheme, which accounts for non-canonical anchor residues. When combined with a comprehensive analysis of the backbone dihedral angles, we describe a framework to explain conformational diversity and introduce D-score as a measure of structural similarity. Using this metric, we find that peptide backbone similarity is allotype-independent and identify 35 discrete peptide configurations which cover the entire seen conformational landscape. Focusing on HLA-A*02:01, we discover strong peptide

sequence trends influenced by distinct backbone features. Finally, we introduce RepPred, an accurate, pan-allelic structural modeling approach for nonamer/HLA complexes, and demonstrate its improved accuracy over existing methods.

There are several important limitations to our study. First, our dataset of X-ray crystal structures is not necessarily a representative sample of in vivo nonamer pHLA structures, and thus, the true distribution of backbone conformations is currently unknown. Additionally, due to the inherent bias in the motivations behind the determination of each crystal structure, our dataset likely contains a high proportion of disease-relevant peptides. Nonetheless, we are confident that we have captured a large fraction of known backbone conformations, as evidenced by our historical analysis of HLA3DB (Fig. 2c). Additionally, as peptide/HLA structures are solved, our automated modeling method will progressively improve due to greater sampling of underrepresented backbone conformations. Second, in utilizing crystallographic structures, we do not account for the dynamic nature of the pHLA complex which has been shown to impact TCR recognition in some cases[71]. For instance, peptide backbones have been observed to exhibit rigid body motions upon TCR binding, and thus could acquire conformations not covered by our discrete backbones[72–74]. Third, RepPred is currently limited to Δ7 nonamers and thus cannot model structures with non-canonical anchors or other peptide lengths. Future efforts can build on RepPred to extend modeling to longer peptides by focusing on the center of the peptide after classifying backbones using the anchor residue-based scheme.

We present HLA3DB (https://hla3db.research.chop.edu), a database of pHLA structures, as an accessible, intuitive resource for the immunology and structural biology community. Beyond providing discrete peptide backbones, its advantage over existing structural databases[42–50] lies in its standardization of structural information. This opens the door for applications including but not limited to serving as a training set for machine learning-based methods for pHLA binding predictions and to ease setup for molecular dynamics simulations.

HLA3DB contains discrete peptide backbone conformations, which were utilized for structural modeling of peptide/HLA-I complexes via RepPred. Going forward, this approach can be improved by training the regression function on data from non-human or class II peptide/MHC structures and incorporating multiple sequence alignment information. Given that the peptide backbone can be accurately modeled, this structural information can be incorporated into existing TCR:pMHC modeling approaches to allow for improved prediction of binding specificity[75–78]. Additionally, RepPred can be utilized to aid the design of chimeric HLA molecules and subsequently identify peptide-centric receptors[79,80]. Finally, leveraging structures in HLA3DB, we envision the identificiation of cross-reactive peptide sequences which, when combined with large-scale structure prediction, can help to finetune immune receptors[10,36–41]. Collectively, HLA3DB provides an atlas of peptide backbones, which RepPred utilizes to traverse from the sequence to the structure space, setting the stage for predicting peptide cross-reactivity in a backbone-focused manner.

## Methods

### HLA3DB curation

The peptide/HLA-I structural dataset (HLA3DB) was curated using a custom Python script (Supplementary Fig. 1). Complexes were identified using the RCSB PDB Search API (v. 2)[51] via a JSON file with the following broad criteria: (i) macromolecular name containing keywords "MHC" or "HLA" or gene name is "HLA-A", "HLA-B", "HLA-C", "HLA-E", or "HLA-G" (ii) source organism is "Homo sapiens", (iii) structure resolution is 3.0 Å or higher, and (iv) structure release date is January 1st, 1988, or later. Using these selection criteria, we obtain a total of 1017 PDB entries (as of April 29th, 2022). Each structure is automatically fetched, filtered and saved into the database using the Bio.PDB Python package from Biopython (v. 1.79)[81]. Next, we check the

PDB file for readability, classify chains into HLA heavy chain, β-2 microglobulin, peptide and other depending on the sequence alignment (with >50% identity) to a reference HLA-A*02:01 heavy chain (consisting of 180 residues from N-terminus), β-2 microglobulin and sequence length (for peptides of length 8 to 10 amino acids). If the PDB file has other chains not within 5 Å from the peptide, the entry is retained for subsequent filtering. Complexes with missing N-terminal residues in the heavy chain are subjected to RosettaRemodel (Rosetta v. 2020.08)[57] where missing one (glycine) or two residues (glycine and serine) from the reference HLA-A*02:01 are modeled. Structures with HLA sequence length of less than 180 residues along the C-termini of the heavy chain are discarded. After trimming the heavy chain up to 180 residues, the structure only contains the peptide/HLA complex. Any structure missing backbone heavy atoms or contains atoms with zero occupancy along the (i) peptide are removed from the database and (ii) heavy chain are reported for manual examination. The PDB entries are then HLA typed by performing pairwise sequence alignment with ~3000 HLA sequences obtained from the IPD-IMGT/HLA Database[82]. The peptide and the heavy chains are renamed, the residues are renumbered, and coordinates are saved in our final structure database. A FASTA file is generated summarizing the dataset with PDB ID, chain name, allele, structure release date, and resolution followed by the sequence of either the HLA or peptide depending on the chain name specified in the previous line. Finally, the dataset was queried for decameric peptides in the Δ7 anchor class and, if appropriate, manual truncation was conducted to create additional nonameric Δ7 peptides, which were named by their original PDB ID followed by "−9".

### Anchor-based classification

The two most distant anchor residues were established by finding the peptide residue with lowest Cα-Cα distance between residue 24 and 123 in the HLA corresponding to the B and F pockets, respectively[7]. Anchor class was determined by subtracting the two most distant anchor residues. The anchor distance was computed by assessing the Cα-Cα distance between the two most distant anchor residues, which define the anchor class. All analysis was completed using custom Python scripts and PyRosetta 4.0 (v. 2020.50)[83].

### Dihedral angle analysis

A free polyglycine nonamer, chosen to eliminate the impact of steric hindrance between HLA residues and peptide side chains, was created using PyMOL[84] Builder (v. 2.5.3) defaulted to an α-helical conformation. Dihedral angles were set using the Python PyMOL package (v. 2.5.3) and iterated through −180° to +180° at 1° intervals for both φ and ψ dihedral angles. For each instance, the anchor distance was computed using the aforementioned method. Dihedral angle pairs that allowed for an anchor distance between 17.5 Å and 20.0 Å were plotted on a Ramachandran plot. Visualization of Ramachandran plots was conducted using a modification of the Ramachandran 0.0.2 Python package.

### D-score structural accuracy metric

The difference between backbone dihedral angles of the same type and residue position was computed using the following equation[55], which was developed to compare antibody loop conformations.

$$D(\theta_1, \theta_2) = 2(1 - \cos(\theta_1 - \theta_2)) \tag{1}$$

This equation accounts for the cyclic nature of dihedral angles, allowing for an accurate difference measurement. We applied this equation to determine the similarity between two Δ7 peptide backbones "A" and "B" for positions "p" as

$$\text{D-score}(A, B) = \sum_{p=4}^{7} \left[ D(\phi_p^A, \phi_p^B) + D(\psi_p^A, \psi_p^B) \right] \tag{2}$$

The D-score criteria states that if A and B have a D-score of ≤1.5, then the two backbones are considered structurally similar or neighbors. For Δ6 and Δ8 peptide backbones, we apply the D-score for P4 to P6 and P4 to P8, respectively, and scale the D-score criteria according to the number of angles.

### Peptide backbone classification using a greedy algorithm

Backbone classification was conducted using a greedy algorithm in Python (v. 3.8.15). Peptide dihedral angles of all Δ7 nonamers were computed, and a binary matrix was created using the D-score criteria such that if two backbones were similar a one was added and otherwise a zero. Then, for each backbone, the number of similar structures were summed. The backbone with the most number of neighbors was noted, and it and its neighbors were removed from the matrix. This process continued iteratively until the matrix was empty.

### Exhaustive structural modeling

Before modeling peptides using pHLA structure templates, we prepared the crystal structures using the FastRelax protocol in Rosetta (v. 2020.08)[85,86]. Then, using a Python script, we iterated through all possible combinations of amino acids between positions 3 to 8 as defined by the HLA-A*02:01 groove bias, which was determined based on a value of 0.0 or lower in a published amino acid similarity matrix derived from experimental data (Supplementary Table 2)[59]. The new peptide sequence was stored in a blueprint file. The anchor residues of P2 and P9 were not included as their sequence was generally conserved across HLA-A*02:01 epitopes. Based on its distance from the center of the peptide, P1 was not included. To model different peptides onto the template pHLA structures, we utilized the RosettaRemodel[57] application from the Rosetta suite of programs. All side chains of the template structure aside from those modeled on the peptide were left in their original poses. The Remodel application automatically scores the modeled structure, and the total score of the complex was utilized. The top 1% of all structures by total Rosetta score were used to establish the peptide sequence space of a given discrete peptide backbone. Next, a position probability matrix (PPM) was created from sequence data with zeroes defaulted to 0.02. The KL divergence was computed using the *rel_entr* function in SciPy (v. 1.7.3)[87] with eluted HLA-A*02:01 and HLA-A*68:01 peptides from the IEDB as a reference.

### Structural modeling of peptide/HLA complexes (RepPred)

**Initial setup.** We apply the greedy algorithm to Δ7 nonamers to obtain discrete peptide backbones. Next, we perform a stability check on the crystal structures of the discrete peptide backbones identified by the greedy algorithm using the Cartesian relax protocol in the Rosetta forcefield. The backbones which move more than a D-score of 1.0 upon relaxation are removed from the template set and the greedy algorithm is reapplied to ensure complete coverage of the conformational space.

**Structural modeling.** For a given template discrete peptide backbone, we set the target amino acid sequence of the nine peptide residues and 180 HLA residues using the *PartialThreadingMover* in PyRosetta (v. 2020.50)[83]. This threaded model is then optimized using the Cartesian relax protocol and four relaxed models are created. If the D-score between the template crystal structure and a given relaxed model is greater than 1.5, that model is removed and not considered during further analysis. We then select the best relaxed model out of the remaining structures by total ref2015 score. Rarely, all four relaxed models may adopt a different backbone after optimization and thus this template backbone would not be considered in further analysis.

**Regression.** For each relaxed model, we compute 129 per residue energy terms of peptide from the ref2015 energy function[29] and these

energy terms are further used for regression analysis. We train a regression function to predict the D-score between the target crystal structure and relaxed model using per residue energy terms of peptides as features. We use the *scikit-learn*[88] (v. 0.0) implementation of Support vector machine regression (SVR) with radial basis function (*sklearn.svm.SVR*). The free parameters of the models, namely the distance parameter epsilon and regularization parameter C, are determined through a grid search hyperparameter scanning. The grid search covers values for epsilon and C in the range $10^{-4}$ to $10^4$ and the best parameter combination is determined using the coefficient of determination ($R^2$) score. The input features of the SVR function are transformed using a uniform Quantile Transformer to avoid nonstandard distributions.

**Benchmarking.** We perform the above three steps to benchmark RepPred. First, we conduct the "Initial setup" step. As a result of this stability check, one discrete peptide backbone was removed and not replaced as it had no neighbors. Thus, we began our modeling by using 33 discrete peptide backbones as templates. Next, we conduct the "Structural modeling" step. We thread the sequence of 260 pHLA complexes (targets) onto the 33 discrete backbone templates, removing structural models generated using a template structure that had a peptide differing from the target peptide at three or fewer residues (homologs). This resulted in 7775 relaxed models (target-template pairs). Finally, we perform the "Regression" step. Here, the SVR function was subjected to a leave-one-out cross validation in which we removed all data corresponding to one target structure from the training dataset and used it as our test case. In the training set only, we removed any target-template pairs with a D-score of greater than 7. For each target, the model with lowest predicted D-score from the SVR is chosen as the best structural model.

The backbone heavy atom and all heavy atom RMSD of the middle of the peptide as well as the backbone heavy atom RMSD of the HLA was computed for A02 targets. For non-A02 targets, RepPred reports a model only if the predicted D-score is less than 2.0. RepPred can model a single target sequence utilizing 136 Xeon 2.10 GHz cores in 30 min.

**Blind testing.** To further confirm the accuracy of RepPred, we performed a blind test. Here, 12 pHLA structures with nonredundant sequences, deposited in the PDB after the cutoff date for HLA3DB, were used. Structural models were created as described previously and the best model was selected using an SVR function trained on 260 nonhomologous pHLA complexes used in the benchmark. Consistent with our benchmarking, the D-score was determined between the best structural model according to RepPred and the crystal structure of the target.

#### Comparison to state-of-the-art methods

In comparing RepPred against six open-source methods for pHLA structural modeling, we maintained all default parameters and followed the appropriate documentation with no change to existing code. For all peptide/HLA-A*02:01 targets, we removed structural homologs for each method except PANDORA. For AF-FT, we removed any targets that were utilized to fine-tune AlphaFold. Some structures could not be processed by existing software. Thus, we report results from 90 structures for GradDock, 99 for APE-Gen, 101 for Keller, 100 for PANDORA, 101 for AlphaFold, and 156 for AF-FT. For GradDock, 3MRE [https://doi.org/10.2210/pdb3MRE/pdb] was used as the template structure as it is the highest resolution HLA-A*02:01 structure in our dataset.

#### Reporting summary

Further information on research design is available in the Nature Portfolio Reporting Summary linked to this article.

## Data availability

The processed pHLA structures are available at https://hla3db.research.chop.edu. Protein structures were obtained from the Protein Data Bank (PDB) using the RCSB PDB Search API (v. 2). A full list of the structures used in this study can be found in the Source Data in the form of PDB IDs. Source data are provided with this paper.

## Code availability

HLA3DB can be accessed via https://hla3db.research.chop.edu. Code used for structural modeling[89] is available on Zenodo via https://doi.org/10.5281/zenodo.8372875.

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

## Acknowledgements

The authors thank Drs. David Haussler, Phil Bradley, and Andrew C. McShan for helpful discussions, and Hailey Wallace from the Rosetta Commons REU program (supported by NSF grant 1659649) for preliminary work during the early stages of the project. This research was supported through grants by NIAID (5R01AI143997), NIGMS (5R35GM125034), and NIDDK (5U01DK112217). This work was delivered as part of the NexTGen team supported by the Cancer Grand Challenges partnership funded by Cancer Research UK (CGCATF-2021/100002) and the National Cancer Institute (CA278687-01) and the Mark Foundation.

## Author contributions

N.G.S. conceived and designed the project. S.G., S.N., and G.L.M. created HLA3DB. S.G. and S.N. performed computational analysis of peptide/HLA crystal structures. S.G. developed the greedy algorithm to identify discrete peptide backbones and conducted exhaustive structural modeling. S.G. and S.K.K. developed and benchmarked RepPred. S.G. and N.G.S. wrote the paper, with feedback from all authors. N.G.S. acquired funding and supervised the project.

## Competing interests

The authors declare no competing interests.
