## [Peer Review File · Nature Communications]

REVIEWER COMMENTS

Reviewer #1 (Remarks to the Author):

The manuscript by Gupta et al focuses on the characterization of peptide backbone structural diversity across available experimentally-determined pHLA-I structures. As part of this effort, the authors produced multiple noteworthy contributions to the field, including (i) an annotated database of experimentally-determined pHLA-I structures (HLA3DB), (ii) a pan-allelic method for pHLA structural modeling (RepPred), and (iii) a new metric for quantifying peptide backbone structural similarity (D-score). In addition, the authors defined a new methodology for classifying peptide backbones based on the distance between anchor residues (anchor class), which led to the significant finding that backbone similarity is allotype-independent (i.e., contrary to common assumptions in the field). However, the rationale for some of the validation steps seems a bit convoluted, the accuracy of predicted side-chains is not discussed, and the authors seem to conflate pHLA structural modeling with T-cell cross-reactivity prediction, despite presenting no results related to the latter topic.

> Comments to the authors:

Major issues:

1) The authors explain that "while there are public available databases that store MHC-I structural data, they do not provide a consistent format needed for further automated analysis." Therefore, in order to perform such automated analysis, the authors created their own annotated database, HLA3DB. Is this database just the means to implement RepPred, or is this database a resource and a contribution on its own? Is my understanding that the authors see HLA3DB as an individual contribution, given the fact that they included HLA3DB in the manuscript's title, and that they implemented a nice webserver to facilitate public access to the database. However, they fail to describe what is the intended use of the database for the general public. The last paragraph of the Discussion must be completely rewritten. It does not present a use case for HLA3DB, and rather merges HLA3DB and RepPred into one approach, for which future improvements and applications are described (see next point). I believe the manuscript can be improved by expanding the Discussion section, and describing each one of the author's contributions (e.g., HLA3DB, D-score, anchor-class, RepPred). Can HLA3DB benefit any user beyond the developers of RepPred? Is there any benefit for the user in relation to data available at PDB, IEDB and IMGT-3Dstructure-DB? Can the anchor class classification facilitate analysis by others? Are the structures cleaned and renumbered in a way that facilitates direct use for Molecular Docking and Molecular Dynamics analysis? Maybe it enables the development/improvement of other modeling methods, the development of better scoring functions for structure-based peptide-HLA binding prediction, or new methods for cross-reactivity prediction, immunogenicity prediction, pHLA stability prediction, etc.

2) The last sentence of the introduction states that "findings enable accurate modeling of peptide/HLA structures at scale, paving the way for accurate prediction of peptide immunogenicity and cross-reactivity". That is a fair claim, and elsewhere in the manuscript the authors cited references to support such claim that structural modeling can be leveraged for immunogenicity prediction and cross-reactivity prediction (please include the references in this sentence as well, line 79). However, the authors go too far at the end of the Discussion, by claiming that HLA3DB can identify cross-reactive peptides; something that was not demonstrated within the manuscript. The last sentence says RepPred uses HLA3DB to offer "an approach for predicting cross-reactivity in a backbone-focused manner". That claim is not consistent with the current manuscript. RepPred is an approach for predicting pHLA structures, which in turn can be used for other analysis, including the prediction of immunogenicity and cross-reactivity. But those downstream applications are not explored here. The authors should either rephrase these claims to be more consistent with presented results, or they should present new results demonstrating the use of HLA3DB and/or RepPred for the task of cross-reactivity prediction. In that case, they should compare their approach with previously proposed methods for cross-reactivity prediction, such as Expitope, iCrossR, sCRAP and MatchTope. On the same point, the last sentence of the Abstract should be revised, since the current manuscript does not provide "a framework for linking conformational diversity with antigen immunogenicity and receptor cross-reactivity".

3) "While low D-score values generally correspond to low RMSD values, there are significant differences between peptide backbones leading to an increased D-score that are not captured by RMSD. Thus, our results suggest that D-score more accurately captures the difference between two peptide conformations". Although the first sentence seems fair, I believe the second is incorrect. Note the first sentence limits the comparison to backbones, while the second implies D-score captures the overall 3D conformation of the peptides, including side chains. Of course, the authors might claim that there are ideal side chain coordinates for a fixed backbone geometry, but they cannot claim D-score captures side chain conformations for any given peptide structure. Maybe a crystal structure or a docking-based model has a peptide side chain in a wrong conformation, despite having the correct backbone geometry. D-scores presented in the manuscript also exclude P1 and Pomega, right? Please make it clear that your discussion is mostly limited to backbone similarity (not global structural similarity).

4) I believe the authors use the term "discrete peptide backbone" to refer to unique backbone geometries. So out of 293 d7 nonamers, they identified 34 "discrete" peptide backbones. But that was not clear from the text. Please revise the text to more clearly define the use of the term "discrete" in this context. See also following comments (5 to 7). The authors are introducing new concepts and a new classification for pHLA structures, so it is key that they clearly define the terms to be used in this new framework, and that they use these terms consistently across the manuscript and Figures.

5) The paragraph starting in line 176 seems to be related to Figure 2, presenting results of the neighbor identification analysis for each discrete d7 (nonamer?) backbone across the entire structural dataset, although Figure 2 is only called later. The authors first call Extended Data Fig. 5A. First, if one of the interesting aspects of the anchor classes is that they are not limited to a particular peptide length, why

was this analysis limited to nonamers? Second, are the results in Extended Data Fig. 5A and 5C related to all anchor classes across the dataset? Are the results in Extended Data Fig. 5B and 5D only restricted to one of the discrete backbones, represented by SAEPVPLQL/HLA-C*05:01? It was not clear which neighbors are been represented in Extended Data Fig. 5D. Why the focus on SAEPVPLQL/HLA-C*05:01? The authors could also include an HLA3DB classification for this backbone geometry, like d7-class66...(see next comment).

6) The authors provide a good description of the proposed anchor classes, d6, d7 and d8. But each of these classes includes a number of discrete backbone geometries, which I believe are referred to as "Backbone classification" in the HLA3DB webserver. There is no description or discussion of such backbone "classes" (?) in the manuscript, other than one reference to "the most common backbone classification (6J1V)", and the algorithm description for backbone classification. But I guess these backbone classes are the same as the "discrete backbone" geometries, as stated in another excerpt on "the most common peptide backbone (PDB ID 6J1V)". This is certainly obvious for the authors, but it is confusing for the unfamiliar reader. Please clarify and use consistent terms across text, figures and the HLA3D database. It is also a bit confusing that the authors use PDB IDs to refer to these different backbone classes, as in the aforementioned excerpts, or in Figure 2D, but they use numbers to identify these classes on HLA3DB. In which order were these numbers assigned? Can we refer to these classes by their numbers? Why many classes don't have a number (NA)? Also, these backbone classes are not shared across anchor classes, right? How many backbone classes we have for each anchor class? (maybe another supplementary figure could help clarify these points).

7) The rationale for the determination of sequence biases is very confusing to follow, and the corresponding text should be fully revised for clarity. The authors ask if there are sequence biases for distinct backbone "classes" (discrete backbones), but don't have enough data to answer this question. Therefore, in order to address this question, the authors model a larger dataset of structures, covering the top 5 most prevalent discrete backbones of HLA-A*0201-restricted d7 nonamer peptides (out of 20 discrete backbones identified). Since one of the contributions of this manuscript is a pHLA modeling tool, RepPred, I would expect RepPred is used for any experiments requiring pHLA modeling. But no; the authors used another unpublished and unverified modeling protocol based on threading of available crystal structures, and rely on Rosetta's scoring function(ref2015) to determine what is "feasible/acceptable". Why is this modeling protocol suited for this analysis? Has it been used before in a similar context? Is the ref2015 scoring function accurate enough for ranking alternative modeled conformations of pHLA structures? Has that been validated before? In addition, rather than modeling all possible amino acids in each position, the authors limited the substitutions based on observed position-specific biases from IEDB data (Sup. Table 1). Therefore, in order to test if there were sequence biases across discrete backbones, the authors predicted structures introducing a sequence bias from IEDB, and relying on a potentially biased scoring function to determine the overall prevalence of different amino acids per peptide position. How can we trust the findings of this experiment?

7b. Then, the authors computed the KL divergence between modeled complexes, and 32,483 experimentally-determined HLA-A2 binders, and describe that "all five distinct backbones exhibited peptide sequence biases". First, what is the meaning of this analysis? Isn't the observation of biases on the modeled structures a demonstration that the modeling method is failing to capture the real sequence biases for HLA-A2-restricted peptides? Second, the authors are comparing the sequence space of the models for the top 5 most prevalent discrete backbones, with a broader dataset of real A2-binders for which they have no information of backbone geometry. So isn't the observed differences only a reflection of the (at least) 15 missing discrete backbone geometries that are possible for A2?

7c. Then, the authors sought to explain these observed biases, by looking closer to the results for 2 out of these 5 discrete backbones. The authors say the D-score captures a "primary (dihedral angle) deviation" at P6, but observe a backbone divergence happening at P3/P4 (Figure 3D). How is that possible? If the backbones are diverging at P3/P4, shouldn't that be reflected on the D-score for one of these two positions? Regardless of what came first and what was a consequent backbone accommodation, the fact is that we have a backbone difference around P4. Please clarify what am I missing here. The authors report a bias favoring tryptophan at P3 for one of the backbones, and a bias favoring small amino acids (e.g., alanine) for the other backbone. Again, how do we know this is a bias introduced by geometric constraints imposed by the discrete backbone conformation, and not by limitations/preferences/biases of the ref2015 scoring function?

8) The same comment on the opaqueness of the rationale applies to the validation of RepPred. First, I think is important to distinguish the components used for modeling, including HLA3DB, RepPred, D-score and the SVR function. The authors say RepPred is "an automated modeling method which utilizes discrete backbones as templates for homology modeling of d7 nonamer epitopes". Rather than choosing the best backbone, RepPred threads the input peptide sequence into all 33 d7 discrete backbones (i.e., 33 templates). The sequence of the MHC is also threaded as needed. This is followed by Cartesian relax protocol with Rosetta. Should this step produce 33 initial models, or even more models per backbone? The relaxation can move the backbone away from its original position, so the D-score between the relaxed model and the original template is used to exclude cases with large backbone deviation. Are these excluded backbones remodeled or lost? In other words, are there backbone geometries that cannot be used for a given peptide sequence, or RepPred always produces an acceptable model for each of the 33 backbone geometries?

8b. Among the models with low D-score to the template, the SVR function is used to rank and select the single best model for that particular input pMHC sequence. So SVR function works like a scoring/ranking function for RepPred. But the explanation of how it works could be improved for clarity. I suggest having a separate Methods subsection talking only about the training of this SVR function, so that we don't mix that with the explanation of the RepPred workflow. The goal of the SVR function is to predict a D-score, to the "real" structure. At the time of training, the authors performed a sort-of "redocking experiment", applying RepPred to a set of known crystal structures of pMHCs. I am assuming that based on the sentence stating "We trained a regression function to predict the D-score between the predicted models

and the corresponding crystal structures of pMHCs using per residue energy terms of peptides as features", but the authors don't specify which crystals are those. Was this done with the set of 247 nonredundant d7 nonamers? By having this explanation as part of the methods on RepPred modeling, it becomes confusing if the authors are talking about D-score with the target complex, or the template complex. So at the time of training, SVR was predicting the D-score to the target structure (which was known). At the time of modeling, SVR will predict the D-score in relation to the unknown real complex. So the model with the lowest SVR-predicted D-score, is the final model produced by RepPred. The SVR function is trained on per-residue energy terms from ref2015. Was ref2015 used before to do pMHC prediction/scoring using per-residue energy terms, or is this an innovation on the use of this scoring function? And the SVR function predicts a D-score value, which is a metric based on dihedral angles. Does it make sense to predict a "geometric deviation" based only on energy terms? Was that needed, or could the authors just have trained a customized version of the ref2015 scoring function using the RepPred models? (ranking based on customized-ref2015 scores, not D-scores, similar to work done in GradDock).

8c. "To assess the efficacy of RepPred, we perform a benchmark on all targets in our dataset against their nonhomologous representatives (Methods)." First, is efficacy the proper term here, or accuracy? Second, this sentence alone does not explain which targets were used for this experiment. More info is available on the Methods, at the paragraph starting at line 447, but again becomes a mixture of information about RepPred and the SVR function. It starts talking about benchmarking RepPred by reproducing 260 HLA-A*0201 complexes; was previously stated that HLA3DB contained 121 A*0201-restricted d7 nonamers, so what complexes are those 260? The next sentence says "in the training set only, we removed any target-template pairs with a D-score of greater than 7"; is this about the previously performed SVR training? There is no training needed at the time of benchmarking RepPred, right? The rest of the paragraph seems to talk about the training and validation of SVR, not RepPred.

8d. I believe lines 276 to 286 were more about SVR validation. Then at 286 the authors state "To validate our modeling approach, we conducted a blind test". So here I believe we are talking about RepPred validation. So were the same 260 complexes used for SVR training and RepPred validation? Was a different dataset used for this "blind test" of RepPred?

8e. Next, the authors compare RepPred with other pHLA modeling tools. Again, the paragraph starts in line 295 describing a "comparison" of RepPred with six existing approaches. This was done in a larger dataset of complexes, although the number of modeled structures is not the same across modeling methods. The authors describe a 76% accuracy for the most prevalent backbone, and 50% accuracy for other backbones. How is success or failure defined in this context? Does it mean that RepPred fails to model half of the targets that do not have the most prevalent backbone? Then, on line 308 the authors say "we validated our method through a blind test on seven non-A2 targets". If this is the validation, what was the previous experiment? What do the authors mean by blind test? Why should we judge the accuracy of these methods on the reproduction of 7 selected complexes, rather than on the basis of hundreds of complexes in the previous experiment?

9) The biggest issue I see in this section is that the authors limit their methods comparison to the analysis of backbone accuracy. They report Backbone-RMSD and D-score for all structures modeled by each method, and identify RepPred as the winner. But if you are comparing a method based on threading over backbone templates from crystal structures, with methods that perform the full sampling of the backbone, it is almost certain that the first will have lower backbone RMSD. But the question is, does the more accurate backbone prediction reflect positively on the all-heavy-atom-RMSD of predicted peptides? Probably it does, but it is not guaranteed given the steps of side-chain relaxation and the biases/limitations of existing scoring functions. In addition, I believe results in Figure 4D are limited to residues between anchors, excluding P1 and Pomega. Therefore, the authors must include in figure 4D the all-heavy-atom-RMSD comparison for these methods. Otherwise the authors cannot claim their method outperforms other methods in the prediction of pHLA geometries.

10) The current paragraph on limitations of this study, in the discussion section, is very good. And I agree that the historical analysis of HLA3DB provides strong evidence that the database already includes a large fraction of discrete backbone geometries (although I am not sure if it has indeed leveled off after 2017, or if it is starting to pick up again after 2021; we will have to wait, since the pandemic had complex impacts on structural data acquisition). However, the authors should extend this discussion to highlight that RepPred is currently limited to d7 anchor class nonamers, despite the fact this class could include other peptide lengths. On the other hand, many of the alternative methods are not limited to nonamers, and could also be used to model peptides from other anchor classes. In the following paragraph authors mention extending the work towards class II MHCs. Does that include dealing with longer peptides, or would still be limited to nonamers? Also, are there any HLA allele limitations for the modeling of d7 nonamers with RepPred, or any allele sequence can be modeled?

Minor issues:

1) Review text regarding typos, punctuation and clarity. There are sentences that are hard to parse, because they contain too many points.

2) The definitions of MHC and HLA are clear in the introduction, but the following sentence of the abstract must be revised, since it conveys that pHLA is the human equivalent of MHC. - "peptide/HLA (pHLA, the human MHC)"

3) This sentence seems to imply that peptide sequence diversity across HLA restrictions can cause disease. I think the authors mean to say that people with different HLAs have different susceptibility to infections and cancers, due to differences in peptide-presentation capacity. Please revise for clarity. - "The large sequence variation...can cause disease susceptibility."

4) When defining the three main docking strategies for pHLA modeling, the authors describe that "fragment-based docking strategies utilize minimization protocols to sample peptide backbone ab initio". I believe the main point of "incremental peptide reconstruction" is the "divide and conquer" approach of dealing with fragments of the ligand, rather than the entire ligand at once. It does not necessarily involve any type of energy minimization, other than the sampling algorithms of the chosen docking tool (e.g., genetic algorithm or other stochastic approach). Please revise.

5) "In terms of HLA peptide binding specificities, the classical MHC-Ia allotypes present in the database cover five out of six known HLA-A and all HLA-B supertypes." I believe MHC-Ia was a typo. Also, please revise the sentence to make it clear the reference is to six HLA-A supertypes, not six known HLA-A (alleles).

6) The HLA3DB webservice has (?) next to each search field, presumably to provide further explanation to the user. This is not working for any of the fields

7) I am not familiar with the format of this Journal, so I was wondering what is the difference between "Extended Data" for each Figure included in the Manuscript, and Supplementary Figures. Aren't the Extended Data Figures part of the Supplementary Information of the manuscript?

8) On line 196 authors mention 293 d7 nonamers, and in line 217 they refer to 247 nonredundant d7 nonamers. So there are 46 redundant d7 nonamer structures in the database? Are redundant structures always consistent in terms of same anchor class and low D-score? Again, in theory we could have different peptide lengths for d7 structures, so why were these analyses limited to nonamers?

9) Regarding lines 259 to 264, "the side chain of P5 cannot contact the backbone nitrogen"; Nitrogen of which position? Also, what is the evidence supporting the claim that this analysis "suggests that long-range interactions could similarly occur between the backbone and HLA groove residues and cause dissimilar pHLA combinations to adopt the same configuration"?

10) The github with the RepPred code is not publicly available yet, so I could not verify that. This github should include a detailed README, a properly documented/commented code, and at least one vignette demonstrating basic usage of RepPred. Since there is no separate github for HLA3DB, maybe this github could also include documentation and vignettes on the use of HLA3D.

10b. How easy is it to install and run RepPred? Rosetta is not the simplest and lightest package to install, so does RepPred installation depend on a previous installation of Rosetta? If so, can the authors provide a Docker image with all required dependencies to facilitate RepPred execution?

10c. On supplementary figure 1 the authors describe a workflow for PDB query, cleaning, processing and storage on HLA3DB. This pHLA structure curation python script(s), or part of them, could be of interest for a broader range of users, interested in molecular docking and molecular dynamics of pHLA complexes. Could this script, or part of it, be also made available through github?

11) On Supplementary table 1, what was the cutoff for the amino acids included in each position? Was it based on yes/no occurrence on IEDB data, or was it based on a particular cutoff of high prevalence in that data?

12) Is it correct to state that the authors didn't find any clues that could be used to predict the most likely backbone geometry for a given target peptide sequence? Instead, they model any target sequence on all possible backbone geometries, and rely on ref2015 and the SVR function to identify the most likely model. Maybe this will be more clear after the revision of the analysis on the sequence biases for the discrete backbone geometries.

13) The modeling with RepPred also includes modeling of the HLA receptor structure, concomitantly with the peptide. Have the authors conducted any analysis of the deviation/accuracy of produced HLA structures in relation to corresponding crystal structures? Maybe this could be provided as supplementary material.

14) If 34 "discrete" peptide backbones were identified for d7 nonamers (line 197), why n=33 in Figure 4A?

15) One of the limitations of previous pHLA modeling methods, particularly backbone template based and fixed terminals, was that they could not model peptides with non-canonical anchors. Is RepPred capable of modeling such peptides?

16) The authors talk about scalability, and did model multiple complexes, but they do not describe the performance of RepPred in terms of computational requirements and time per complex modeled. Also, the largest modeling in the manuscript was not performed with RepPred. Can RepPred be applied to model thousands of complexes?

Reviewer #2 (Remarks to the Author):

This paper presents a useful database and analysis of human Class I MHC peptide complexes, including a conformational analysis of the peptide antigen within the binding groove of HLA proteins. It also presents an algorithm and program for predicting the structures of HLA-Class-I/peptide complexes that is competitive with deep learning methods.

There are some shortcomings in the paper that could be addressed. The data as presented seem incomplete in several respects and there are some issues with presentation and the website. The Github code is not available which makes it impossible to try the structure prediction method to see if it is easy to use and whether it produces reasonable results.

Here are more detailed comments in roughly decreasing order of importance.

1. It would make sense to cluster the delta6 (N=15) and delta8 (N=75) classes. The delta6 group may or may not form clusters with the D-score=1.5 cutoff (or some scaled value for 6 dihedrals instead of 8 used for delta-7). I would like to see Ramachandran maps for each class for all residues (one row of maps per class with 7, 8, 9 residues for delta6, delta7, delta8 respectively. After clustering, I would like to see maps for each cluster over a certain size (e.g., 10 structures). Examples can be given in the main text, with complete data in the Supplemental information. Sequence logos of each cluster of experimental structures should be provided. For an example, see Figure 3 in Kelow paper mentioned below.

Without the Ramachandran maps for the unclustered data and for the clusters, it's not clear that the clusters are separated by regions of low density or whether they are divisions of continuous density. It would be useful in Supplemental data to show clusters at different D-score cutoffs to demonstrate that the D-score cutoff used is justified (this was done with DBSCAN parameters in the paper on kinases listed below).

2. It might be interesting to include all of residues P2-P8 in the D-score calculation since the phi, psi of residues 3 and 8 are involved in the relative displacement of residues 2 and 9. In the delta7 peptides. psi of residue 2 and phi of residue 9 may also play a role. If there are no significant deviations of the P2,P3,P8,P9, adding them to the D-score equation will not change the clustering results (depending on the cutoff).

3. It's not clear if the cluster numbers (used as names) will be stable. At least the larger ones could be named or at least given descriptions based on the Ramachandran maps of the clustered residues. See the paper on kinases listed below (Modi and Dunbrack). For instance, the cluster in Extended Fig 5 could

be APPP. The North paper used A,B,D,P,L,G to label regions of the Ramachandran map (G was later relabeled “E”, which is more apt). It depends on whether each cluster can be assigned a unique label by the region of the Ramachandran map occupied by most members of the cluster. In any case, for the clustering of delta7 loops a table of counts, Ramachandran dihedral angle means and standard deviations, and Ramachandran region designations should be given.

4. Some entries have multiple copies of the HLA-peptide complex in the asymmetric unit. These should be analyzed separately, and reported by chainIDs of the MHC protein and the peptide (e.g. 3VXP). There are at least 150 such entries. This would provide more data for clustering, although two chains in the same asymmetric unit are likely to be very similar.

5. The downloadable data could be improved. Filenames for PDB structures can contain about HLA allele, delta class, and conformational cluster. Data should really be in mmCIF format not PDB format in 2023. The PDB files should have sequence records and other meta data, not just coordinates.

6. The webpage search results could be improved. Make everything clickable - like the allele on the PDB page, so you can go to a page with all structures of that allele. When showing the results of a search, show the whole result not just the first 10. Since the database is not large, the result is never more than 100 structures and would load quickly. Having to click through 8 pages for 75 structures is unnecessarily annoying.

7. Captions are sometimes uninformative. E.g. Extended Data Figure 6. These are three structures of the HLA-A2 protein with structural modeling results. But the caption does not mention HLA-A2. It's not clear why panel D is 5HHQ and 6VR1 and what their dihedral angle differences have to do with the rest of the figure.

8. In Extended Data Figure 5, the example of HLA-C-5*01 is shown with 12 members of the same conformational cluster (peptide sequence SAEPVPLQL). But on the HLA3DB website, this cluster has only 2 members (cluster #39).

9. The paper states: “While there are publicly available databases that store MHC-I structural data, they do not provide a consistent format needed for further automated analysis [36-44].” This citation does not make sense. Literally none of these papers are databases that store MHC-I structural data. There must be other HLA/MHC databases which contain structural information, even if they do not present conformational clustering as the present paper does. The literature on these databases should be reviewed in the introduction and/or compared in the Discussion.

10. I really don't like having to mention this, but it seems necessary. The paper does not accurately credit work that it depends on. The conformational clustering depends on the "D-score", which the authors claim to have "established" (Results section) without any citation. In the Methods section, they cite North et al for the distance equation between two dihedral angles:

$$D(\theta_1, \theta_2) = 2(1 - \cos(\theta_1 - \theta_2)).$$

Then say that they "extended this equation" to establish the D-score:

$$D\text{-score}(A,B) = \text{sum over residues } (p=4,5,6,7) \text{ of } D(\phi_p_A, \phi_p_B) + D(\psi_p_A, \psi_p_B)$$

This is exactly Eq. 2 in the North et al paper where it was called "D". The authors neither "extended" the single-dihedral equation for the first time nor "established" the D-score. Since the paper depends on the D-score, the D-score should be properly attributed and cited in the main body of the paper.

More importantly, we used the same methods in other papers, which the authors might examine on how to present the kind of data they discuss in this paper. The same equation is presented in each of them. The Kincore website (<http://dunbrack.fccc.edu/kincore>) might be a useful example for the HLA3DB website.

Modi and Dunbrack, PNAS, 2019 (on kinases). <https://www.pnas.org/doi/full/10.1073/pnas.1814279116>

Kelow, Adolf-Bryfogle, and Dunbrack, mAbs, 2020 (on the antibody CDR L4 and H4 loops). <https://www.tandfonline.com/doi/full/10.1080/19420862.2020.1840005>

Shapovalov, Vucetic, Dunbrack, PLOS Comp Biol, 2019 (beta turns)

<https://doi.org/10.1371/journal.pcbi.1006844>

Parker, Meyer, Golemis, Dunbrack, Cancer Research 2022 (on RAS)

<https://doi.org/10.1158/0008-5472.CAN-22-0804>

Kelow, Faezov, Xu, Parker, Adolf-Bryfogle, Dunbrack, biorxiv doi, 2022 (updated clustering of antibody CDRs). <https://doi.org/10.1101/2022.10.12.511988>

REVIEWER COMMENTS

Reviewer #1 (Remarks to the Author):

The manuscript by Gupta et al focuses on the characterization of peptide backbone structural diversity across available experimentally-determined pHLA-I structures. As part of this effort, the authors produced multiple noteworthy contributions to the field, including (i) an annotated database of experimentally-determined pHLA-I structures (HLA3DB), (ii) a pan-allelic method for pHLA structural modeling (RepPred), and (iii) a new metric for quantifying peptide backbone structural similarity (D-score). In addition, the authors defined a new methodology for classifying peptide backbones based on the distance between anchor residues (anchor class), which led to the significant finding that backbone similarity is allotype-independent (i.e., contrary to common assumptions in the field). However, the rationale for some of the validation steps seems a bit convoluted, the accuracy of predicted side-chains is not discussed, and the authors seem to conflate pHLA structural modeling with T-cell cross-reactivity prediction, despite presenting no results related to the latter topic.

We thank the reviewer for their positive appraisal of our work, and for helpful suggestions to strengthen our conclusions.

> Comments to the authors:

Major issues:

1) The authors explain that "while there are public available databases that store MHC-I structural data, they do not provide a consistent format needed for further automated analysis." Therefore, in order to perform such automated analysis, the authors created their own annotated database, HLA3DB. Is this database just the means to implement RepPred, or is this database a resource and a contribution on its own? Is my understanding that the authors see HLA3DB as an individual contribution, given the fact that they included HLA3DB in the manuscript's title, and that they implemented a nice webserver to facilitate public access to the database. However, they fail to describe what is the intended use of the database for the general public. The last paragraph of the Discussion must be completely rewritten. It does not present a use case for HLA3DB, and rather merges HLA3DB and RepPred into one approach, for which future improvements and applications are described (see next point). I believe the manuscript can be improved by expanding the Discussion session, and describing each one of the author's contributions (e.g., HLA3DB, D-score, anchor-class, RepPred). Can HLA3DB benefit any user beyond the developers of RepPred? Is there any benefit for the user in relation to data available at PDB, IEDB and IMGT-3Dstructure-DB? Can the anchor class classification facilitate analysis by others? Are the structures cleaned and renumbered in a way that facilitates direct use for Molecular Docking and Molecular Dynamics analysis? Maybe it enables the development/improvement of other modeling methods, the development of better scoring functions for structure-based peptide-HLA binding prediction, or new methods for cross-reactivity prediction, immunogenicity prediction, pHLA stability prediction, etc.

We agree with the reviewer that HLA3DB represents a significant contribution on its own and can benefit a broad range of users. Our database marks an improvement over existing websites such as IMGT-3Dstructure-DB as it provides standardized structures rather than a copy of the coordinates from the PDB. As the reviewer mentions, this will enable applications including but not limited to molecular dynamics simulations, development of future pHLA modeling methods, and cross-reactivity predictions. Also, the anchor class classification scheme can facilitate the analysis of peptide/MHC-II structures and for longer peptides presented by MHC-I. Finally, we anticipate RepPred being used for cross-reactivity predictions. We have reworked the Discussion section and included this suggestion by the reviewer on lines 330 to 334.

2) The last sentence of the introduction states that "findings enable accurate modeling of peptide/HLA structures at scale, paving the way for accurate prediction of peptide immunogenicity and cross-reactivity". That is a fair claim, and elsewhere in the manuscript the authors cited references to support such claim that structural modeling can be leveraged for immunogenicity prediction and cross-reactivity prediction (please include the references in this sentence as well, line 79). However, the authors go too far at the end of the Discussion, by claiming that HLA3DB can identify cross-reactive peptides; something that was not demonstrated within the manuscript. The last sentence says RepPred uses HLA3DB to offer "an approach for predicting cross-reactivity in a backbone-focused manner". That claim is not consistent with the current manuscript. RepPred is an approach for predicting pHLA structures, which in turn can be used for other analysis, including the prediction of immunogenicity and cross-reactivity. But those downstream applications are not explored here. The authors should either rephrase these claims to be more consistent with presented results, or they should present new results demonstrating the use of HLA3DB and/or RepPred for the task of cross-reactivity prediction. In that case, they should compare their approach with previously proposed methods for cross-reactivity prediction, such as Expitope, iCrossR, sCRAP and MatchTope. On the same point, the last sentence of the Abstract should be revised, since the current manuscript does not provide "a framework for linking conformational diversity with antigen immunogenicity and receptor cross-reactivity".

We thank the reviewer for their comments on the applicability of our work. We have included the appropriate references at the end of the Introduction section on line 73. We have rephrased the sentences in the Abstract and Discussion to be more consistent with the presented results on line 22 and lines 343 to 345, respectively.

3) "While low D-score values generally correspond to low RMSD values, there are significant differences between peptide backbones leading to an increased D-score that are not captured by RMSD. Thus, our results suggest that D-score more accurately captures the difference between two peptide conformations". Although the first sentence seems fair, I believe the second is incorrect. Note the first sentence limits the comparison to backbones, while the second implies D-score captures the overall 3D conformation of the peptides, including side chains. Of course, the authors might claim that there are ideal side chain coordinates for a fixed backbone geometry, but they

cannot claim D-score captures side chain conformations for any given peptide structure. Maybe a crystal structure or a docking-based model has a peptide side chain in a wrong conformation, despite having the correct backbone geometry. D-scores presented in the manuscript also exclude P1 and Pomega, right? Please make it clear that your discussion is mostly limited to backbone similarity (not global structural similarity).

We thank the reviewer for this comment which enhances the clarity of our manuscript. We have modified the text to include this recommendation on lines 154 to 155 and lines 163 to 165 where we note that the D-score is specifically used to measure backbone similarity. Indeed, the D-score excludes P1 and P Ω as well as P2, P3, and P8 for a $\Delta 7$ peptide. Thus, it only includes the center of the peptide as delineated in the formula in the Methods section on line 392 to 394.

4) I believe the authors use the term "discrete peptide backbone" to refer to unique backbone geometries. So out of 293 d7 nonamers, they identified 34 "discrete" peptide backbones. But that was not clear from the text. Please revise the text to more clearly define the use of the term "discrete" in this context. See also following comments (5 to 7). The authors are introducing new concepts and a new classification for pHLA structures, so it is key that they clearly define the terms to be used in this new framework, and that they use these terms consistently across the manuscript and Figures.

We apologize for any confusion in the terminology used in the text and have clarified it throughout the manuscript. We specifically define a discrete peptide backbone as "the structure with the most neighbors" during the explanation of the greedy algorithm on line 182.

5) The paragraph starting in line 176 seems to be related to Figure 2, presenting results of the neighbor identification analysis for each discrete d7 (nonamer?) backbone across the entire structural dataset, although Figure 2 is only called later. The authors first call Extended Data Fig. 5A. First, if one of the interesting aspects of the anchor classes is that they are not limited to a particular peptide length, why was this analysis limited to nonamers? Second, are the results in Extended Data Fig. 5A and 5C related to all anchor classes across the dataset? Are the results in Extended Data Fig. 5B and 5D only restricted to one of the discrete backbones, represented by SAEPVPLQL/HLA-C*05:01? It was not clear which neighbors are been represented in Extended Data Fig. 5D. Why the focus on SAEPVPLQL/HLA-C*05:01? The authors could also include an HLA3DB classification for this backbone geometry, like d7-class66...(see next comment).

We thank the reviewer for this insightful idea and have modified our analysis to include all peptides in an anchor class irrespective of peptide length. Additionally, we have clarified our commentary of Extended Data Fig. 5 to delineate that SAEPVPLQL/HLA-C*05:01 (PDB ID 5VGD) is not a discrete peptide backbone. Instead, it is an example of how peptides bound to different allotypes can have similar backbone conformations. We present this as a precursor to our more comprehensive analysis

using the greedy algorithm to identify discrete peptide backbones. We have clarified this on lines 171 to 172.

6) The authors provide a good description of the proposed anchor classes, d6, d7 and d8. But each of these classes includes a number of discrete backbone geometries, which I believe are referred to as "Backbone classification" in the HLA3DB webserver. There is no description or discussion of such backbone "classes" (?) in the manuscript, other than one reference to "the most common backbone classification (6J1V)", and the algorithm description for backbone classification. But I guess these backbone classes are the same as the "discrete backbone" geometries, as stated in another excerpt on "the most common peptide backbone (PDB ID 6J1V)". This is certainly obvious for the authors, but it is confusing for the unfamiliar reader. Please clarify and use consistent terms across text, figures and the HLA3D database. It is also a bit confusing that the authors use PDB IDs to refer to these different backbone classes, as in the aforementioned excerpts, or in Figure 2D, but they use numbers to identify these classes on HLA3DB. In which order were these numbers assigned? Can we refer to these classes by their numbers? Why many classes don't have a number (NA)? Also, these backbone classes are not shared across anchor classes, right? How many backbone classes we have for each anchor class? (maybe another supplementary figure could help clarify these points).

We apologize for the confusion between the terminology used in the manuscript and on the HLA3DB webserver. We have altered the wording to be consistent across all aspects of our work. Specifically, we name a discrete backbone first by its anchor class and then by a backbone number. This number is assigned in the order of how many neighbors it has. For instance, the most common $\Delta 7$ backbone would be $\Delta 7-1$. In the manuscript, we also include the corresponding PDB ID of the discrete peptide backbone. An example of this terminology can be found on line 201 ($\Delta 7-1_6J1V$) and we have also included it in a table of the top classes for each anchor class (Supplementary Table 1).

Previously, on the HLA3DB webserver, many structures had a class named "NA" as we had restricted our analysis to $\Delta 7$ nonamers. However, we have expanded our work to include all peptides in our database and thus all structures have a valid classification on the webserver. The reviewer is correct that backbone classes are not shared across anchor classes. Instead, each backbone class is tied to a given anchor class i.e., $\Delta 7-3$. For the $\Delta 6$, $\Delta 7$, and $\Delta 8$ anchor classes, there are 3, 35, and 34 peptide backbone classes. This finding is delineated in the text on lines 187 to 188 and we have included examples of the top backbones in the Supplementary Figure 3, 4, and 5 and in Supplementary Table 1.

7) The rationale for the determination of sequence biases is very confusing to follow, and the corresponding text should be fully revised for clarity. The authors ask if there are sequence biases for distinct backbone "classes" (discrete backbones), but don't have enough data to answer this question. Therefore, in order to address this question, the authors model a larger dataset of structures, covering the top 5 most prevalent discrete backbones of HLA-A*0201-restricted d7 nonamer peptides (out of 20 discrete

backbones identified). Since one of the contributions of this manuscript is a pHLA modeling tool, RepPred, I would expect RepPred is used for any experiments requiring pHLA modeling. But no; the authors used another unpublished and unverified modeling protocol based on threading of available crystal structures, and rely on Rosetta's scoring function(ref2015) to determine what is "feasible/acceptable". Why is this modeling protocol suited for this analysis? Has it been used before in a similar context? Is the ref2015 scoring function accurate enough for ranking alternative modeled conformations of pHLA structures? Has that been validated before? In addition, rather than modeling all possible amino acids in each position, the authors limited the substitutions based on observed position-specific biases from IEDB data (Sup. Table 1). Therefore, in order to test if there were sequence biases across discrete backbones, the authors predicted structures introducing a sequence bias from IEDB, and relying on a potentially biased scoring function to determine the overall prevalence of different amino acids per peptide position. How can we trust the findings of this experiment?

We apologize for any confusion in the text. Our hypothesis was to determine if the stereochemical constraints that are specific to different peptide backbones result in differences in peptide sequences. We used the RosettaRemodel protocol to test this hypothesis, as it enabled us to conduct a fixed backbone analysis. RepPred was not suited for this analysis since it considers multiple peptide backbones, and therefore these are not fixed in our method.

This modeling protocol, while implemented here using RosettaRemodel, has been used in similar manners as described in the RosettaDesign web server publication¹. Several papers²⁻⁶ using the ref2015 scoring function have extensively tested the validity of this protocol through experimental structure determination. Furthermore, we do not explicitly rank modeled conformations, and instead select the top 1% of the distribution.

Therefore, while the ref2015 scoring function may not be accurate enough to differentiate individual models, published data suggests that selecting from a distribution of structures can yield accurate results. Nonetheless, we recognize that all scoring functions have biases, and we acknowledge this limitation of our analysis on lines 222 to 224 when we note that ref2015 has struggled with electrostatic interactions.

We have clarified the text to delineate how the sequence biases from IEDB data were generated on lines 216 to 218. Specifically, a positional scanning combinatorial library was generated⁷ and used to build a peptide:MHC binding energy covariance (PMBEC) matrix⁸ which was used as a Bayesian prior for a Stabilized Matrix Method (SMM)⁹, an algorithm used to predict peptide/MHC binding. Since peptide sequence biases were based on experimental data and the scoring function has been experimentally validated, we maintain that the findings of our experiment can be trusted, in the sense that it shows that different peptide backbones arise from specific peptide sequence patterns.

7b. Then, the authors computed the KL divergence between modeled complexes, and 32,483 experimentally-determined HLA-A2 binders, and describe that "all five distinct backbones exhibited peptide sequence biases". First, what is the meaning of this analysis? Isn't the observation of biases on the modeled structures a demonstration that the modeling method is failing to capture the real sequence biases for HLA-A2-restricted peptides? Second, the authors are comparing the sequence space of the

models for the top 5 most prevalent discrete backbones, with a broader dataset of real A2-binders for which they have no information of backbone geometry. So isn't the observed differences only a reflection of the (at least) 15 missing discrete backbone geometries that are possible for A2?

We apologize for any confusion. We used the KL divergence as a quantitative measure of peptide sequence bias. Additionally, we confirm the values by visual inspection of the sequence logos in Fig. 3C and Extended Data Fig. 6A, B, and C. The observation of biases is not related to the real sequence bias for HLA-A*02:01 restricted peptides as this was not the intention of our experiment. We sought to determine if discrete peptide backbones had differences in the distribution of sequences that can adopt these structures.

Indeed, the observed differences reflect the 15 missing discrete backbones. Our analysis is an indication that discrete backbones can have peptide sequence preferences. If we performed the same procedure on the other 15 backbones, we would expect to capture most of the peptides which can be presented by HLA-A*02:01. However, this analysis lies outside of the scope of our work and would be computationally expensive. Thus, we focused on the five most common backbones.

7c. Then, the authors sought to explain these observed biases, by looking closer to the results for 2 out of these 5 discrete backbones. The authors say the D-score captures a "primary (dihedral angle) deviation" at P6, but observe a backbone divergence happening at P3/P4 (Figure 3D). How is that possible? If the backbones are diverging at P3/P4, shouldn't that be reflected on the D-score for one of these two positions? Regardless of what came first and what was a consequent backbone accommodation, the fact is that we have a backbone difference around P4. Please clarify what am I missing here. The authors report a bias favoring tryptophan at P3 for one of the backbones, and a bias favoring small amino acids (e.g., alanine) for the other backbone. Again, how do we know this is a bias introduced by geometric constraints imposed by the discrete backbone conformation, and not by limitations/preferences/biases of the ref2015 scoring function?

We apologize for the typo in our analysis. Indeed, the dihedral angle differences are at P5 and P6 and the backbone divergence is also at these positions. We have corrected this on lines 238 to 240.

While the ref2015 scoring function has struggled with capturing subtle changes involving electrostatic interactions¹⁰, it has demonstrated its ability to capture steric hindrance in published studies¹⁻⁶. We have noted this limitation in this section on lines 222 to 224.

8) The same comment on the opaqueness of the rationale applies to the validation of RepPred. First, I think is important to distinguish the components used for modeling, including HLA3DB, RepPred, D-score and the SVR function. The authors say RepPred is "an automated modeling method which utilizes discrete backbones as templates for homology modeling of d7 nonamer epitopes". Rather than choosing the best backbone, RepPred threads the input peptide sequence into all 33 d7 discrete backbones (i.e., 33

templates). The sequence of the MHC is also threaded as needed. This is followed by Cartesian relax protocol with Rosetta. Should this step produce 33 initial models, or even more models per backbone? The relaxation can move the backbone away from its original position, so the D-score between the relaxed model and the original template is used to exclude cases with large backbone deviation. Are these excluded backbones remodeled or lost? In other words, are there backbone geometries that cannot be used for a given peptide sequence, or RepPred always produces an acceptable model for each of the 33 backbone geometries?

We thank the reviewer for their suggestion on the organization of the RepPred Methods section. We have modified the Methods section to create subsections describing RepPred's development. We state that after relaxation, four models are produced for each template backbone. We remove models that have a large D-score between the relaxed model and the original template and select the best relaxed model out of the remaining structures by ref2015 score. On occasion, all four models may have a large D-score so a backbone may be lost. Thus, RepPred does not necessarily produce 33 relaxed models from all templates in the end. We have clarified this on lines 430 to 434.

8b. Among the models with low D-score to the template, the SVR function is used to rank and select the single best model for that particular input pMHC sequence. So SVR function works like a scoring/ranking function for RepPred. But the explanation of how it works could be improved for clarity. I suggest having a separate Methods subsection talking only about the training of this SVR function, so that we don't mix that with the explanation of the RepPred workflow. The goal of the SVR function is to predict a D-score, to the "real" structure. At the time of training, the authors performed a sort-of "redocking experiment", applying RepPred to a set of known crystal structures of pMHCs. I am assuming that based on the sentence stating "We trained a regression function to predict the D-score between the predicted models and the corresponding crystal structures of pMHCs using per residue energy terms of peptides as features", but the authors don't specify which crystals are those. Was this done with the set of 247 nonredundant d7 nonamers? By having this explanation as part of the methods on RepPred modeling, it becomes confusing if the authors are talking about D-score with the target complex, or the template complex. So at the time of training, SVR was predicting the D-score to the target structure (which was known). At the time of modeling, SVR will predict the D-score in relation to the unknown real complex. So the model with the lowest SVR-predicted D-score, is the final model produced by RepPred. The SVR function is trained on per-residue energy terms from ref2015. Was ref2015 used before to do pMHC prediction/scoring using per-residue energy terms, or is this an innovation on the use of this scoring function? And the SVR function predicts a D-score value, which is a metric based on dihedral angles. Does it make sense to predict a "geometric deviation" based only on energy terms? Was that needed, or could the authors just have trained a customized version of the ref2015 scoring function using the RepPred models? (ranking based on customized-ref2015 scores, not D-scores, similar to work done in GradDock).

We apologize for any confusion in the Methods section that addresses RepPred. We have followed the reviewer's suggestion and included a dedicated Methods subsection on the regression function that has been clarified according to the reviewer's insightful comments which can be found on lines 435 to 444.

We have modified the text in the Methods section to provide a general, three step process and a separate subsection for the benchmarking and blind testing of our method. To answer the reviewer's question, we trained the regression function on the D-score between the relaxed model and the target crystal structure. For the benchmark, this was done for 260 pHLA complexes across 33 discrete backbone templates, not including homologs, which equates to 7,775 target-template pairs.

We used an SVR function to predict pMHC structures on the basis of per-residue energy terms as this was described in a recent paper from Keller and colleagues¹¹. Here, they use kinematic loop modeling to create structural models and test several regression functions to predict the heavy atom RMSD between the model and crystal structure. We have cited them extensively throughout our paper and use their best regression function ("radSVR") in our protocol. RepPred uses structural modeling, building on the knowledge of discrete peptide backbone conformations, instead of loop modeling and predicts D-score instead of RMSD as it is a better metric of comparing peptide backbones. We demonstrate that RepPred outperforms their method ("Keller"). We thank the reviewer for their suggestion on training a customized version of the ref2015 scoring function. While this approach is feasible, given the accuracy of GradDock in comparison to Keller, we chose to build on the latter. Additionally, GradDock's approach¹² has been tested rigorously in their paper and we did not aim to train another energy function and instead chose to reweight the energy terms as this method was simpler to implement and had shown considerable success in recent literature. Nonetheless, we agree with the reviewer's suggestion and will certainly explore this idea in future iterations of RepPred. Notwithstanding, our method shows significant improvements over the state-of-the-art approaches.

8c. "To assess the efficacy of RepPred, we perform a benchmark on all targets in our dataset against their nonhomologous representatives (Methods)." First, is efficacy the proper term here, or accuracy? Second, this sentence alone does not explain which targets were used for this experiment. More info is available on the Methods, at the paragraph starting at line 447, but again becomes a mixture of information about RepPred and the SVR function. It starts talking about benchmarking RepPred by reproducing 260 HLA-A*0201 complexes; was previously stated that HLA3DB contained 121 A*0201-restricted d7 nonamers, so what complexes are those 260? The next sentence says "in the training set only, we removed any target-template pairs with a D-score of greater than 7"; is this about the previously performed SVR training? There is no training needed at the time of benchmarking RepPred, right? The rest of the paragraph seems to talk about the training and validation of SVR, not RepPred.

We apologize for any confusion. We have replaced the word "efficacy" for "accuracy" on line 262.

Additionally, we have reorganized the Methods section of RepPred and clarified the text in the "Benchmarking" subsection. We apologize for the typo in the Methods section. The 260 pHLA complexes are all $\Delta 7$ nonamers that are not the discrete peptide backbones and they come from all alleles not just HLA-A*02:01 (line 449). During the benchmarking of RepPred, the SVR function is subjected to a leave-one-out cross-validation where all data corresponding to a target structure is removed from the training set and this target structure is considered our test case. Then, to train the SVR function, we remove any target-template pairs with a D-score greater than 7. Training is required during the benchmarking of RepPred but is performed such that our training and testing data is split responsibly. This is clarified on lines 452 to 456.

8d. I believe lines 276 to 286 were more about SVR validation. Then at 286 the authors state "To validate our modeling approach, we conducted a blind test". So here I believe we are talking about RepPred validation. So were the same 260 complexes used for SVR training and RepPred validation? Was a different dataset used for this "blind test" of RepPred?

We apologize for any confusion. The text on line 274 and line 297 have been revised to make it clear that the blind test is used for *further* validation along with the benchmark. To perform the blind test, we use a separate dataset of 12 pHLA structures and use a SVR regression trained on the 260 pHLA complexes used in benchmarking. The procedure for blind testing is clarified in a dedicated subsection of the Methods on line 461 to 466.

8e. Next, the authors compare RepPred with other pHLA modeling tools. Again, the paragraph starts in line 295 describing a "comparison" of RepPred with six existing approaches. This was done in a larger dataset of complexes, although the number of modeled structures is not the same across modeling methods. The authors describe a 76% accuracy for the most prevalent backbone, and 50% accuracy for other backbones. How is success or failure defined in this context? Does it mean that RepPred fails to model half of the targets that do not have the most prevalent backbone? Then, on line 308 the authors say "we validated our method through a blind test on seven non-A2 targets". If this is the validation, what was the previous experiment? What do the authors mean by blind test? Why should we judge the accuracy of these methods on the reproduction of 7 selected complexes, rather than on the basis of hundreds of complexes in the previous experiment?

We apologize for any confusion. The number of modeled structures varies across approaches because not all sequences could result in a structure for existing software. While we had 110 available $\Delta 7$ nonamer/HLA-A*02:01 structures in HLA3DB, only 96 structures for GradDock, 108 for APE-Gen, and 109 structures for PANDORA could be processed. This is stated in the Methods on lines 472 to 474. Additionally, for AF-FT, we removed targets that the method was trained on as stated on lines 470 to 471.

We define success on line 267 to 268.

While it is true that RepPred fails to model half of targets which do not have the most prevalent backbone, we note that this accuracy is higher than its competitor, AF-FT which is 38% accurate.

We revised the text on line 274 and line 297 to clarify that the blind test was performed for *further* validation. A blind test means that we have not seen the pHLA structures before as they were deposited after the cutoff data for HLA3DB. We judge the accuracy of the methods using both a leave-one-out cross-validation method and using a blind test. We provide additional information in the “Blind testing” subsection in the Methods on lines 461 to 466.

9) The biggest issue I see in this section is that the authors limit their methods comparison to the analysis of backbone accuracy. They report Backbone-RMSD and D-score for all structures modeled by each method, and identify RepPred as the winner. But if you are comparing a method based on threading over backbone templates from crystal structures, with methods that perform the full sampling of the backbone, it is almost certain that the first will have lower backbone RMSD. But the question is, does the more accurate backbone prediction reflect positively on the all-heavy-atom-RMSD of predicted peptides? Probably it does, but it is not guaranteed given the steps of side-chain relaxation and the biases/limitations of existing scoring functions. In addition, I believe results in Figure 4D are limited to residues between anchors, excluding P1 and Pomega. Therefore, the authors must include in figure 4D the all-heavy-atom-RMSD comparison for these methods. Otherwise the authors cannot claim their method outperforms other methods in the prediction of pHLA geometries.

We thank the reviewer for this suggestion. The plot of all heavy-atom RMSD of the peptide shows that RepPred performs competitively with existing methods, including AF-FineTune, which we state on lines 285 to 286. This supports the claim that an accurate backbone prediction generally translates to proper side chain placement. In future versions of RepPred we plan to build on the backbone prediction to improve side chain accuracy by implementing more comprehensive protein side-chain packing tools such as AttnPacker¹³, DLPacker¹⁴, FASPR¹⁵, and SCWRL4¹⁶. To maintain the focus of this manuscript on the prediction of the peptide backbone conformation, we have included a comparison of the methods using all-heavy-atom RMSD in Extended Data Fig. 7C rather than in Figure 4D.

Extended Data Fig. 7C. Boxplots showing the distribution of peptide all-heavy-atom RMSD of RepPred and six state-of-the-art methods, sorted by publication date. Whiskers extend to the furthest values that lie within the 75th and 25th percentile value ± 1.5 times the interquartile range and outliers are shown in black circles. An orange dashed line is at an RMSD of 1.5 and the percentage of models under this threshold for each method is listed above the respective boxplot.

10) The current paragraph on limitations of this study, in the discussion section, is very good. And I agree that the historical analysis of HLA3DB provides strong evidence that the database already includes a large fraction of discrete backbone geometries (although I am not sure if it has indeed leveled off after 2017, or if it is starting to pick up again after 2021; we will have to wait, since the pandemic had complex impacts on structural data acquisition). However, the authors should extend this discussion to highlight that RepPred is currently limited to d7 anchor class nonamers, despite the fact this class could include other peptide lengths. On the other hand, many of the alternative methods are not limited to nonamers, and could also be used to model peptides from other anchor classes. In the following paragraph authors mention extending the work towards class II MHCs. Does that include dealing with longer peptides, or would still be limited to nonamers? Also, are there any HLA allele limitations for the modeling of d7 nonamers with RepPred, or any allele sequence can be modeled?

We thank the reviewer for their suggestion. We have extended our discussion to include the limitation that RepPred can presently only model $\Delta 7$ nonamers on line 326 to 329. For class II MHCs, we are currently preparing a manuscript that addresses longer peptides using a similar homology modeling protocol. RepPred does not have any HLA allele limitations as it employs homology modeling to generate structures. For this reason, we claim on line 314 that our method is “pan-allelic”.

Minor issues:

1) Review text regarding typos, punctuation and clarity. There are sentences that are hard to parse, because they contain too many points.

We apologize for these errors. We have reworked significant portions of the manuscript to enhance clarity.

2) The definitions of MHC and HLA are clear in the introduction, but the following sentence of the abstract must be revised, since it conveys that pHLA is the human equivalent of MHC. - "peptide/HLA (pHLA, the human MHC)"

We apologize for this error. We have clarified the text of the abstract on line 15.

3) This sentence seems to imply that peptide sequence diversity across HLA restrictions can cause disease. I think the authors mean to say that people with different HLAs have different susceptibility to infections and cancers, due to differences in peptide-presentation capacity. Please revise for clarity. - "The large sequence variation...can cause disease susceptibility."

We apologize for the lack of clarity. We revised this on line 29 to 30.

4) When defining the three main docking strategies for pHLA modeling, the authors describe that "fragment-based docking strategies utilize minimization protocols to sample peptide backbone ab initio". I believe the main point of "incremental peptide reconstruction" is the "divide and conquer" approach of dealing with fragments of the ligand, rather than the entire ligand at once. It does not necessarily involve any type of energy minimization, other than the sampling algorithms of the chosen docking tool (e.g., genetic algorithm or other stochastic approach). Please revise.

We apologize for this error. We revised this on line 51 to 52.

5) "In terms of HLA peptide binding specificities, the classical MHC-Ia allotypes present in the database cover five out of six known HLA-A and all HLA-B supertypes." I believe MHC-Ia was a typo. Also, please revise the sentence to make it clear the reference is to six HLA-A supertypes, not six known HLA-A (alleles).

We apologize for this typo. We revised this on line 86.

6) The HLA3DB webserver has (?) next to each search field, presumably to provide further explanation to the user. This is not working for any of the fields

We thank the reviewer for visiting our webserver. Upon testing this feature, we believe it is functioning properly and kindly request the reviewer to retry it.

7) I am not familiar with the format of this Journal, so I was wondering what is the difference between "Extended Data" for each Figure included in the Manuscript, and

Supplementary Figures. Aren't the Extended Data Figures part of the Supplementary Information of the manuscript?

We thank the reviewer for their question.

From the Nature website: "Extended Data is an integral part of the paper and only data that directly contribute to the main message should be presented. These figures will be integrated into the full-text HTML version of your paper and will be appended to the online PDF."

Meanwhile, "Supplementary Information is material that is essential background to the study but which it is not practical to include in the PDF version of the paper (for example, video files, large data sets and calculations)."

8) On line 196 authors mention 293 d7 nonamers, and in line 217 they refer to 247 nonredundant d7 nonamers. So there are 46 redundant d7 nonamer structures in the database? Are redundant structures always consistent in terms of same anchor class and low D-score? Again, in theory we could have different peptide lengths for d7 structures, so why were these analysis limited to nonamers?

We thank the reviewer for this observation. Among $\Delta 7$ nonamers, we found 46 structures that have at least one other structure with the same peptide sequence. Analyzing their peptide backbones revealed that 45% of the time, their conformation was different (D-score > 1.5). We did not include this finding as it detracts from the main focus of the paper but have provided a graph for the reviewer.

We have included all $\Delta 7$ peptides regardless of peptide length in our discrete backbone analysis but removed the octamer and decamer peptides when investigating peptide sequence biases as the different peptide lengths would have increased the complexity of the experiment.

As per the reviewer's prior suggestions, we have expanded our analysis for peptides in the $\Delta 7$ class to be peptide length independent.

9) Regarding lines 259 to 264, "the side chain of P5 cannot contact the backbone nitrogen"; Nitrogen of which position? Also, what is the evidence supporting the claim that this analysis "suggests that long-range interactions could similarly occur between the backbone and HLA groove residues and cause dissimilar pHLA combinations to adopt the same configuration"?

We apologize for this unsupported claim. We have clarified which position of nitrogen we are referring to on line 249 and have removed our comments on long-range interactions.

10) The github with the RepPred code is not publicly available yet, so I could not verify that. This github should include a detailed README, a properly documented/commented code, and at least one vignette demonstrating basic usage of RepPred. Since there is no separate github for HLA3DB, maybe this github could also include documentation and vignettes on the use of HLA3D.

We apologize for this error. We have made the code for HLA3DB and RepPred readily accessible via GitHub (<https://www.github.com/titaniumsg/RepPred>) and included documentation and an example.

10b. How easy is to install and run RepPred? Rosetta is not the simplest and lightest package to install, so does RepPred installation depends on a previous installation of Rosetta? If so, can the authors provide a Docker image with all required dependencies to facilitate RepPred execution?

We thank the reviewer for their comments on the accessibility of our method to the general scientific community. As Rosetta installation requires a license and documentation on installation is widely accessible, we chose not to provide a Docker image. Instead, we provide a YAML file which can be used to generate the proper Anaconda environment to run HLA3DB and RepPred after downloading the code from GitHub.

10c. On supplementary figure 1 the authors describe a workflow for PDB query, cleaning, processing and storage on HLA3DB. This pHLA structure curation python script(s), or part of them, could be of interest for a broader range of users, interested in molecular docking and molecular dynamics of pHLA complexes. Could this script, or part of it, be also made available through github?

We thank the reviewer for their interest. This script has been made readily available on GitHub (<https://www.github.com/titaniumsg/RepPred>) with proper documentation.

11) On Supplementary table 1, what was the cutoff for the amino acids included in each position? Was it based on yes/no occurrence on IEDB data, or was it based on a particular cutoff of high prevalence in that data?

We thank the reviewer for this observation. We used a previously published matrix that displays amino acid similarities derived from experimental binding affinity measurements⁸. We selected amino acids that were less than 0.0 in this matrix for HLA-A*02:01 and included proline due to its restraining effects on the peptide. We have updated this in the Methods section on lines 406 to 409.

12) Is it correct to state that the authors didn't find any clues that could be used to predict the most likely backbone geometry for a given target peptide sequence? Instead, they model any target sequence on all possible backbone geometries, and rely on ref2015 and the SVR function to identify the most likely model. Maybe this will be more clear after the revision of the analysis on the sequence biases for the discrete backbone geometries.

We thank the reviewer for their suggestion. While we did find that distinct backbone conformations had sequence biases, these trends were not strong enough to aid our structure prediction effort.

13) The modeling with RepPred also includes modeling of the HLA receptor structure, concomitantly with the peptide. Have the authors conducted any analysis of the deviation/accuracy of produced HLA structures in relation to corresponding crystal structures? Maybe this could be provided as supplementary material.

We thank the reviewer for their suggestion. We have provided a distribution of the backbone heavy atom RMSD of the HLA between the final RepPred model and the target crystal structure for all targets in Extended Data Fig. 7A. As expected, the accuracy is sub-angstrom for 96% of targets as the HLA is generally conserved irrespective of peptide conformation. We have included a comment in the Results section on line 265 to 266.

Extended Data Fig. 7A. Distribution of HLA backbone heavy atom RMSD between the RepPred model and the corresponding target crystal structure for all targets.

14) If 34 "discrete" peptide backbones were identified for d7 nonamers (line 197), why n=33 in Figure 4A?

We thank the reviewer for this observation. We have clarified the difference in the number of discrete peptide backbones in the "Benchmarking" subsection of the Methods on lines 430 to 434. As a result of the stability check on the backbone templates, one discrete peptide backbone was removed as the D-score between the relaxed structure and the crystal structure was greater than 1.0. Since this backbone did not have any neighbors, it was not replaced.

15) One of the limitations of previous pHLA modeling methods, particularly backbone template based and fixed terminals, was that they could not model peptides with non-canonical anchors. Is RepPred capable of modeling such peptides?

We thank the reviewer for their suggestion. RepPred, in its current implementation, assumes that the input peptide sequence is a $\Delta 7$ nonamer. However, $\Delta 7$ peptides of length 8 or 10 amino acids could theoretically be modeled by only considering the peptide backbone conformation between the anchor residues. As per the reviewer's prior suggestion, we have included this limitation in the Discussion on lines 326 to 329.

16) The authors talk about scalability, and did model multiple complexes, but they do not describe the performance of RepPred in terms of computational requirements and time per complex modeled. Also, the largest modeling in the manuscript was not performed with RepPred. Can RepPred be applied to model thousands of complexes?

On line 459 to 460 of the Methods section we write, "RepPred can model a single target sequence utilizing 136 Xeon 2.10 GHz cores in 30 minutes." Given sufficient computational resources, we maintain that RepPred can be used to model thousands of complexes at scale. To determine peptide sequence biases, modeling was conducted with the intention of maintaining a fixed backbone and thus RepPred was not utilized as it can alter the conformation. Additionally, the required number of computations, nearly 4 million models, would be too computationally expensive for RepPred.

Reviewer #2 (Remarks to the Author):

This paper presents a useful database and analysis of human Class I MHC peptide complexes, including a conformational analysis of the peptide antigen within the binding groove of HLA proteins. It also presents an algorithm and program for predicting the structures of HLA-Class-I/peptide complexes that is competitive with deep learning methods.

There are some shortcomings in the paper that could be addressed. The data as presented seem incomplete in several respects and there are some issues with presentation and the website. The Github code is not available which makes it impossible to try the structure prediction method to see if it is easy to use and whether it produces reasonable results.

We thank the reviewer for their positive appraisal of our work. Below, we provide a detailed response to the very insightful comments raised.

Here are more detailed comments in roughly decreasing order of importance.

1. It would make sense to cluster the delta6 (N=15) and delta8 (N=75) classes. The delta6 group may or may not form clusters with the D-score=1.5 cutoff (or some scaled value for 6 dihedrals instead of 8 used for delta-7). I would like to see Ramachandran maps for each class for all residues (one row of maps per class with 7, 8, 9 residues for delta6, delta7, delta8 respectively). After clustering, I would like to see maps for each cluster over a certain size (e.g., 10 structures). Examples can be given in the main text, with complete data in the Supplemental information. Sequence logos of each cluster of experimental structures should be provided. For an example, see Figure 3 in Kellow paper mentioned below.

Without the Ramachandran maps for the unclustered data and for the clusters, it's not clear that the clusters are separated by regions of low density or whether they are divisions of continuous density. It would be useful in Supplemental data to show clusters at different D-score cutoffs to demonstrate that the D-score cutoff used is justified (this was done with DBSCAN parameters in the paper on kinases listed below).

We thank the reviewer for their suggestion. We have provided all residue Ramachandran plots for all three anchor classes in Extended Data Figure 3 ($\Delta 7$) and Supplementary Figure 2 ($\Delta 6$ and $\Delta 8$).

Extended Data Figure 3. Analysis of $\Delta 7$ peptides in HLA3DB. (A) General Ramachandran plots of all $\Delta 7$ peptides ($n = 303$). (B) Percentage of structures with a PPII deviation at a given dihedral angle and position.

Supplementary Figure 2. Analysis of $\Delta 6$ and $\Delta 8$ peptides in HLA3DB. (A) General Ramachandran plots of all $\Delta 6$ peptides ($n = 15$). (B) General Ramachandran plots of all $\Delta 8$ peptides ($n = 75$). Plots are colored and formatted identically to (A).

Additionally, after classification of peptides across these three anchor classes, we have provided additional information for the top 3 backbone classes for $\Delta 7$ (Supplementary Figure 4) and the top class for $\Delta 6$ (Supplementary Figure 3) and $\Delta 8$ (Supplementary Figure 5), according to the number of neighbors. The classes are divisions of continuous density since our greedy algorithm is not meant for clustering. Instead, it finds the minimum set of discrete peptide backbones which can represent the conformational landscape corresponding to peptides of a single anchor class. This is delineated in the text on line 176 to 178.

Supplementary Figure 3. Additional information of the most common backbone classes among the $\Delta 6$ peptides. (A) Ramachandran plot of the most common backbone class (n = 13). (B) Sequence logo of peptides shown in (A).

Supplementary Figure 4. Additional information of the three most common backbone classes among the $\Delta 7$ peptides. (A) Ramachandran plot of the most common backbone class (n = 105). (B) Sequence logo of peptides shown in (A). (C) Ramachandran plot of

the second most common backbone class (n = 37). (D) Sequence logo of peptides shown in (C). (E) Ramachandran plot of the third most common backbone class (n = 28). (F) Sequence logo of peptides shown in (E).

Supplementary Figure 5. Additional information of the most common backbone classes among the $\Delta 8$ peptides. (A) Ramachandran plot of the most common backbone class (n = 13). (B) Sequence logo of peptides shown in (A).

We chose the D-score cutoff of 1.5 based on structural similarity rather than forming classes. As expected, increasing the D-score cutoff decreases the number of discrete peptide backbones. However, this is at the expense of covering the entire conformational landscape. While clustering methods such as DBSCAN have parameters which require finetuning, our greedy algorithm uses a fixed D-score cutoff.

2. It might be interesting to include all of residues P2-P8 in the D-score calculation since the phi, psi of residues 3 and 8 are involved in the relative displacement of residues 2 and 9. In the delta7 peptides, psi of residue 2 and phi of residue 9 may also play a role. If there are no significant deviations of the P2,P3,P8,P9, adding them to the D-score equation will not change the clustering results (depending on the cutoff).

We thank the reviewer for their suggestion. While the Ramachandran plots show no significant deviations in the P2, P3, P8, and P9 dihedral angles, we find that introducing them to the D-score calculation introduces noise. Plotting the original D-score (P4 to P7) against this modified D-score (P2 to P9), we find that the median difference is 0.54. However, when applying this linear correction, we observe that one-third of backbone comparisons are misclassified. Thus, to preserve structural similarity, we retained the original D-score (P4 to P7).

3. It's not clear if the cluster numbers (used as names) will be stable. At least the larger ones could be named or at least given descriptions based on the Ramachandran maps of the clustered residues. See the paper on kinases listed below (Modi and Dunbrack). For instance, the cluster in Extended Fig 5 could be APPP. The North paper used

A,B,D,P,L,G to label regions of the Ramachandran map (G was later relabeled “E”, which is more apt). It depends on whether each cluster can be assigned a unique label by the region of the Ramachandran map occupied by most members of the cluster. In any case, for the clustering of delta7 loops a table of counts, Ramachandran dihedral angle means and standard deviations, and Ramachandran region designations should be given.

We thank the reviewer for their suggestion. We have included a new table (Supplementary Table 1) that includes all relevant information present in Table 3 of the North et al paper. As expected, each class can be assigned to more than one region of the Ramachandran plot.

Class	# of Structures	Peptide Conformation	Median D-score
Δ 6-1_1AGE	13	PBB,PPB	0.14
Δ 6-2_4F7T	1	BBB	0
Δ 6-3_1E28	1	PPP	0
Δ 7-1_6J1V	105	ABBB,ABBP,APBB	1.03
Δ 7-2_7M8U	37	APBB,APPB,DPPB	0.92
Δ 7-3_6UZM	28	PPBP,PPPP	0.43
Δ 7-4_1UXS	27	PBAP,PBDP,PPAB,PPDB,PPDP	1.02
Δ 7-5_5IEK	19	ABPB,ABPP,ADPB,DBPB,DDPP	0.64
Δ 8-1_3MRR	13	ABLBB,ABLPB,DBLBB,DBLPB	0.65
Δ 8-2_5FDW	8	APPPP	0.68
Δ 8-3_5C0I	6	BABPP	0.73
Δ 8-4_3DX7	5	BBPLB,BBPLP,BPPLP,PPPLP	1.09
Δ 8-5_3OXR	4	PPAPP,PPDPP	0.32

Supplementary Table 1. Peptide conformations are included if they are present in at least 10% of structures in a given class and labeled according to previously established regions of the Ramachandran plot¹⁷. The median D-score was calculating by median of the D-score of all members of the class to the discrete PDB ID.

4. Some entries have multiple copies of the HLA-peptide complex in the asymmetric unit. These should be analyzed separately, and reported by chainIDs of the MHC protein and the peptide (e.g. 3VXP). There are at least 150 such entries. This would provide more data for clustering, although two chains in the same asymmetric unit are likely to be very similar.

We thank the reviewer for this suggestion. We reconfigured our curation script to capture pHLA complexes in the asymmetric unit and found that the majority of structures were similar in backbone conformation to their selected counterpart. Due to

the redundant nature of this dataset, we chose not to include this finding in this paper but have provided the structures in the HLA3DB webserver.

5. The downloadable data could be improved. Filenames for PDB structures can contain about HLA allele, delta class, and conformational cluster. Data should really be in mmCIF format not PDB format in 2023. The PDB files should have sequence records and other meta data, not just coordinates.

We thank the reviewer for this suggestion. We have included mmCIF formatted data in the download alongside PDB data. We have also included sequence records and other metadata in the PDB file. Since each download contains a CSV file indicating the HLA allele, anchor class, backbone classification, and further information, we chose to maintain the filenames for structures as is.

6. The webpage search results could be improved. Make everything clickable - like the allele on the PDB page, so you can go to a page with all structures of that allele. When showing the results of a search, show the whole result not just the first 10. Since the database is not large, the result is never more than 100 structures and would load quickly. Having to click through 8 pages for 75 structures is unnecessarily annoying.

We thank the reviewer for improving the accessibility of our website. We have increased the “clickability” of our search page. Additionally, we have included the option to display all structures on one page by using the “Show ‘n’ entries” feature above the queried table.

7. Captions are sometimes uninformative. E.g. Extended Data Figure 6. These are three structures of the HLA-A2 protein with structural modeling results. But the caption does not mention HLA-A2. It’s not clear why panel D is 5HHQ and 6VR1 and what their dihedral angle differences have to do with the rest of the figure.

We apologize for the confusion. The dihedral angle difference shown in Extended Data Fig 6D accompanies the comparison of the backbones of 5HHQ and 6VR1 in Figure 3D. Overall, Extended Data Fig. 6 is used to further support the claims made in Figure 3.

8. In Extended Data Figure 5, the example of HLA-C-5*01 is shown with 12 members of the same conformational cluster (peptide sequence SAEPVPLQL). But on the HLA3DB website, this cluster has only 2 members (cluster #39).

We apologize for any confusion. We have clarified our commentary of Extended Data Fig. 5 to delineate that SAEPVPLQL/HLA-C*05:01 (PDB ID 5VGD) is not a discrete peptide backbone. Instead, it is an example of how peptides bound to different allotypes can give rise to similar backbone conformations. We present this as a precursor to our more comprehensive analysis using the greedy algorithm to identify discrete peptide backbones. We have clarified this on line 171 to 172.

9. The paper states: “While there are publicly available databases that store MHC-I structural data, they do not provide a consistent format needed for further automated analysis [36-44].” This citation does not make sense. Literally none of these papers are databases that store MHC-I structural data. There must be other HLA/MHC databases which contain structural information, even if they do not present conformational clustering as the present paper does. The literature on these databases should be reviewed in the introduction and/or compared in the Discussion.

We apologize for this error in our references. We have updated the entries to refer to the correct papers. Additionally, we delineate the advantage of HLA3DB over these databases in the Discussion on line 331 to 333. Namely, our database marks an improvement over existing websites such as IMGT-3Dstructure-DB as it provides standardized structures rather than a copy of the coordinates from the PDB. This will enable applications including but not limited to molecular dynamics simulations, development of future pHLA modeling methods, and cross-reactivity predictions

10. I really don't like having to mention this, but it seems necessary. The paper does not accurately credit work that it depends on. The conformational clustering depends on the “D-score”, which the authors claim to have “established” (Results section) without any citation. In the Methods section, they cite North et al for the distance equation between two dihedral angles:

$$D(\theta_1, \theta_2) = 2(1 - \cos(\theta_1 - \theta_2)).$$

Then say that they “extended this equation” to establish the D-score:

$$D\text{-score}(A,B) = \sum \text{over residues } (p=4,5,6,7) \text{ of } D(\phi_p_A, \phi_p_B) + D(\psi_p_A, \psi_p_B)$$

This is exactly Eq. 2 in the North et al paper where it was called “D”. The authors neither “extended” the single-dihedral equation for the first time nor “established” the D-score. Since the paper depends on the D-score, the D-score should be properly attributed and cited in the main body of the paper.

More importantly, we used the same methods in other papers, which the authors might examine on how to present the kind of data they discuss in this paper. The same equation is presented in each of them. The Kincore website (<http://dunbrack.fccc.edu/kincore>) might be a useful example for the HLA3DB website.

Modi and Dunbrack, PNAS, 2019 (on kinases). <https://www.pnas.org/doi/full/10.1073/pnas.1814279116>

Kelow, Adolf-Bryfogle, and Dunbrack, mAbs, 2020 (on the antibody CDR L4 and H4 loops). <https://www.tandfonline.com/doi/full/10.1080/19420862.2020.1840005>

Shapovalov, Vucetic, Dunbrack, PLOS Comp Biol, 2019 (beta turns) <https://doi.org/10.1371/journal.pcbi.1006844>

Parker, Meyer, Golemis, Dunbrack, Cancer Research 2022 (on RAS) <https://doi.org/10.1158/0008-5472.CAN-22-0804>

Kelow, Faezov, Xu, Parker, Adolf-Bryfogle, Dunbrack, biorxiv doi:, 2022 (updated clustering of antibody CDRs). <https://doi.org/10.1101/2022.10.12.511988>

We sincerely apologize for this transgression. The original papers by Dunbrack and colleagues were indeed a primary motivating factor for our work, and our adaptation of this structural similarity metric and analysis methods are by no means novel to or established by our study (in fact, our use of the term D-score is based on the name Dunbrack). Unfortunately, due to a lack of diligence from our part, these foundational studies were not referenced in the final, submitted version of our manuscript. We have revised the wording in the Results and Methods section and have cited the North et al paper in the Results section on line 148. Additionally, we have clarified that our work builds on this research by citing all publications in the Discussion section on line 307.

References

1. Liu, Y. & Kuhlman, B. RosettaDesign server for protein design. *Nucleic Acids Res* 34, W235–W238 (2006).
2. Dantas, G., Kuhlman, B., Callender, D., Wong, M. & Baker, D. A Large Scale Test of Computational Protein Design: Folding and Stability of Nine Completely Redesigned Globular Proteins. *Journal of Molecular Biology* 332, 449–460 (2003).
3. Korkegian, A., Black, M. E., Baker, D. & Stoddard, B. L. Computational thermostabilization of an enzyme. *Science* 308, 857–860 (2005).
4. Kuhlman, B. & Baker, D. Native protein sequences are close to optimal for their structures. *Proc Natl Acad Sci U S A* 97, 10383–10388 (2000).
5. Ambroggio, X. I. & Kuhlman, B. Computational design of a single amino acid sequence that can switch between two distinct protein folds. *J Am Chem Soc* 128, 1154–1161 (2006).
6. Kuhlman, B. *et al.* Design of a Novel Globular Protein Fold with Atomic-Level Accuracy. *Science* 302, 1364–1368 (2003).
7. Sidney, J. *et al.* Quantitative peptide binding motifs for 19 human and mouse MHC class I molecules derived using positional scanning combinatorial peptide libraries. *Immunome Res* 4, 2 (2008).
8. Kim, Y., Sidney, J., Pinilla, C., Sette, A. & Peters, B. Derivation of an amino acid similarity matrix for peptide: MHC binding and its application as a Bayesian prior. *BMC Bioinformatics* 10, 394 (2009).

9. Peters, B. & Sette, A. Generating quantitative models describing the sequence specificity of biological processes with the stabilized matrix method. *BMC Bioinformatics* 6, 132 (2005).
10. Das, R. Four Small Puzzles That Rosetta Doesn't Solve. *PLOS ONE* 6, e20044 (2011).
11. Keller, G. L. J., Weiss, L. I. & Baker, B. M. Physicochemical Heuristics for Identifying High Fidelity, Near-Native Structural Models of Peptide/MHC Complexes. *Frontiers in Immunology* 13, (2022).
12. Kyeong, H.-H., Choi, Y. & Kim, H.-S. GradDock: rapid simulation and tailored ranking functions for peptide-MHC Class I docking. *Bioinformatics* 34, 469–476 (2018).
13. McPartlon, M. & Xu, J. An end-to-end deep learning method for protein side-chain packing and inverse folding. *Proceedings of the National Academy of Sciences* 120, e2216438120 (2023).
14. Misiura, M., Shroff, R., Thyer, R. & Kolomeisky, A. B. DLPacker: Deep learning for prediction of amino acid side chain conformations in proteins. *Proteins: Structure, Function, and Bioinformatics* 90, 1278–1290 (2022).
15. Huang, X., Pearce, R. & Zhang, Y. FASPR: an open-source tool for fast and accurate protein side-chain packing. *Bioinformatics* 36, 3758–3765 (2020).
16. Krivov, G. G., Shapovalov, M. V. & Dunbrack, R. L. Improved prediction of protein side-chain conformations with SCWRL4. *Proteins* 77, 778–795 (2009).
17. North, B., Lehmann, A. & Dunbrack, R. L. A new clustering of antibody CDR loop conformations. *J Mol Biol* 406, 228–256 (2011).

REVIEWERS' COMMENTS

Reviewer #1 (Remarks to the Author):

I am satisfied with the revised version produced by the authors. All previously identified issues/questions have been addressed.

Reviewer #2 (Remarks to the Author):

The authors have responded well to my previous comments about their manuscript. The website is much better now that it is more "clickable." The paper presents a useful tool for understanding HLA/MHC structural biology.

Reviewer #1 (Remarks to the Author):

I am satisfied with the revised version produced by the authors. All previously identified issues/questions have been addressed.

We thank the reviewer for their constructive feedback during the review process.

Reviewer #2 (Remarks to the Author):

The authors have responded well to my previous comments about their manuscript. The website is much better now that it is more "clickable." The paper presents a useful tool for understanding HLA/MHC structural biology.

We thank the reviewer for their constructive feedback during the review process and comment on the impact of our work.